# Replication collisions induced by de-repressed S-phase transcription are connected with malignant transformation of adult stem cells

Ting Zhang[1], Carsten Künne [1,2], Dong Ding[1], Stefan Günther [1,3], Xinyue Guo[1], Yonggang Zhou[1], Xuejun Yuan [1] ✉ & Thomas Braun [1,4,5] ✉

Transcription replication collisions (TRCs) constitute a major intrinsic source of genome instability but conclusive evidence for a causal role of TRCs in tumor initiation is missing. We discover that lack of the H4K20-dimethyltransferase KMT5B (also known as SUV4-20H1) in muscle stem cells de-represses S-phase transcription by increasing H4K20me1 levels, which induces TRCs and aberrant R-loops in oncogenic genes. The resulting replication stress and aberrant mitosis activate ATR-RPA32-P53 signaling, promoting cellular senescence, which turns into rapid rhabdomyosarcoma formation when p53 is absent. Inhibition of S-phase transcription ameliorates TRCs and formation of R-loops in *Kmt5b*-deficient MuSCs, validating the crucial role of H4K20me1-dependent, tightly controlled S-phase transcription for preventing collision errors. Low *KMT5B* expression is prevalent in human sarcomas and associated with tumor recurrence, suggesting a common function of *KMT5B* in sarcoma formation. The study uncovers decisive functions of KMT5B for maintaining genome stability by repressing S-phase transcription via control of H4K20me1 levels.

Genomic instability is a hallmark of most malignancies. Tumor cells are assumed to be progeny of genetically instable single cells, which continue to acquire further abnormalities over time[1]. Since mutations in DNA repair genes are infrequent in non-hereditary cancers, genomic instability most likely originates from replication stress, defined as slowing or stalling of replication forks[2]. Overexpression of oncogenes such as *Ras*, *Myc*, *Cdc25A* or *cyclin E* are known to inflict replication stress, which has been attributed, at least in part, to collisions between transcription and replication[3]. Such collisions lead to stalled DNA replication forks and accumulation of RNA:DNA hybrids (R-loops)[4,5]. Stalled DNA replication forks are prone to collapse, provoking DNA

double strand breaks and chromosomal rearrangements in human cancers[4]. Cells have developed several strategies to tolerate or prevent collisions between both machineries, such as co-orientation of transcription and replication to avoid head-on collision as well as spatio-temporal separation of either process[6,7]. Failure to separate transcription from replication will inevitably lead to collisions. Despite these insights, it is not clear whether TRCs serve as a causative driver for tumor initiation. Likewise, the molecular mechanisms causing TRCs in different pathophysiological settings are poorly explored.

Replication and transcription are both affected by the chromatin structure, determining which process is initiated and how fast it

[1]Department of Cardiac Development and Remodeling, Max-Planck-Institute for Heart and Lung Research, Bad Nauheim, Germany. [2]Bioinformatics Core Unit (BCU), Max Planck Institute for Heart and Lung Research, Bad Nauheim, Germany. [3]CPI Deep Sequencing Platform, Max Planck Institute for Heart and Lung Research, Bad Nauheim, Germany. [4]German Center for Cardiovascular Research (DZHK), Rhine Main, Germany. [5]German Center for Lung Research (DZL), Giessen, Germany. ✉e-mail: xuejun.yuan@mpi-bn.mpg.de; thomas.braun@mpi-bn.mpg.de

proceeds. Chromatin changes are prevalent in tumor stem cells, suggesting that alterations in the epigenome might be a driving force for tumorigenesis[8]. Information about the exact roles of epigenetic modifiers for coordinating transcription and DNA replication is still critically missing. One well-known cell cycle dependent histone modification is the methylation of lysine 20 on histone H4. Activity of the H4K20-monomethyltransferase KMT5A/PR-SET7/SET8 and presence of its product H4K20me1 oscillates during the cell cycle with highest levels during mitosis, a decline during late G1, and lowest levels in S-phase. Maintenance of H4K20me1 levels was found to be critical for proper cell cycle progression, DNA replication, mitosis, and transcription[9]. Furthermore, recent evidence indicates that H4K20me1 promotes transcription via directly facilitating opening of chromatin[10]. In agreement with these findings, we previously demonstrated that KMT5B, which converts H4K20me1 to H4K20me2, controls high order heterochromatin formation and maintains quiescence of MuSCs in vivo by repressing transcription of key myogenic factors[11].

MuSCs, also known as muscle satellite cells, are essential for adult muscle homeostasis and regeneration[12,13]. MuSCs are predominantly quiescent in resting skeletal muscles, but become activate and proliferate upon muscle injury[14]. Similar to other adult stem cells, MuSCs act as cells of origin for cancer formation, which happens after loss of *p53* and constant activation and proliferation of MuSCs as in mdx mice, leading to formation of rhabdomyosarcomas (RMS)[15,16]. Interestingly, MuSCs in mdx mice show a substantial decline of *Kmt5b* expression, but a potential impact on genome stability has not been interrogated so far[11]. RMS is the most common pediatric malignant sarcoma, displaying features of myogenic differentiation such as the expression of *MyoD*, *MyoG*, and *desmin*[17–19]. Furthermore, RMSs show a high frequency of copy number alterations and/or allelic imbalances[20,21], suggesting that chromosomal rearrangements or aneuploidy triggered by replication stress is causally involved in rhabdomyosarcoma formation.

Here, we demonstrate that KMT5B coordinates transcription with replication by conversion of H4K20me1 to H4K20me2, which is instrumental for suppression of transcriptional activity during S-phase, prevention of TRCs and aberrant R-loop formation in gene-bodies. Replication stress induced by loss of *Kmt5b* and subsequent massive accumulation of DNA damage activate the ATR-P53 pathway, preventing further proliferation and inducing senescence in MuSC. Inactivation of *p53* prevents senescence and allows *Kmt5b*-deficient MuSCs to pass the cell-cycle checkpoint, leading to malignant transformation and rhabdomyosarcoma formation. Our study demonstrates a causative role of TRCs and increased R-loop formation for tumor development and identifies *Kmt5b* as a potent tumor suppressor.

## Results

### Kmt5b prevents replication stress and DNA damage in MuSCs

In a previous study we found that inactivation of *Kmt5b* causes precocious activation of quiescent MuSCs[11]. To further investigate the role of *Kmt5b* for MuSC expansion, we monitored proliferation of MuSCs from *Pax7^CreERT^/Kmt5b^+/+^* mice (= Ctrl) and *Pax7^CreERT^/Kmt5b^flox/flox^* (= *Kmt5b^sKO^*) using time lapse imaging. Loss of *Kmt5b* (Supplementary Fig. 1a) reduced expansion of MuSCs but strongly increased intensity of γH2X staining in EdU+ cells and the percentage of 53BP1+/EdU+ cells, indicating enhanced DNA damage during S-phase (Fig. 1a, b; Supplementary Fig. 1b). Importantly, a marked increase of 53BP1 nuclear bodies (NB) was apparent in the G1-phase of *Kmt5b^sKO^* MuSCs, suggesting failure to complete DNA replication during the last cell cycle (Supplementary Fig. 1c). In addition, DNA fiber spreading assay revealed reduced DNA fiber progression and increased replication fork stalling in *Kmt5b^sKO^* MuSCs, demonstrating that *Kmt5b* depletion causes DNA replication stress (Fig. 1c, d). We also observed substantially increased numbers of 53BP1+/PAX7+ and γH2X+/PAX7+ cells in regenerating TA muscles from *Kmt5b^sKO^*

mice 7-days after cardiotoxin (CTX)-induced muscle injury (Supplementary Fig. 1d, e).

We next investigated whether loss of *Kmt5b* causes chromosomal instability. We detected an increased frequency of aberrant mitoses, including centrosome amplification-induced multipolar mitotic spindles and anaphase bridging in *Kmt5b^sKO^* MuSC, leading to accumulation of aneuploid and micronuclei-containing MuSCs (Fig. 1e, f; Supplementary Fig. 1f, g). Since DNA under-replication and damage induce P53-dependent cell cycle withdrawal and senescence, we monitored expression levels of P53 and its targets. Expression levels of P53 and *p21* were strongly elevated in *Kmt5b^sKO^* MuSCs, concurrent with increased accumulation of γH2AX (Fig.1g, h). FACS anlaysis revealed that the percentage of cells in either G0/G1- or S-phase was not significantly altered, although *Kmt5b* depletion resulted in a slight increase of cells in G2/M-phase, probably caused by aberrant mitoses following replication stress in mutant cells (Supplementary Fig. 1h). *Kmt5b^sKO^* MuSCs were enlarged and flattened, exhibiting a senescence-like morphology and enhanced activity of senescence-associated β-galactosidase (SA-β-GAL), whereas no significant induction of apoptosis was observed (Fig. 1i; Supplementary Fig. 1i). Taken together, we concluded that loss of *Kmt5b* causes DNA damage, replication stress, and genome instability in proliferating *Kmt5b^sKO^* MuSCs both in vitro and in vivo.

### Kmt5b limits accumulation of H4K20me1 and Pol II S2P on gene bodies in MuSCs

Next, we explored the consequences of a loss of *Kmt5b* for histone modifications in MuSC, in which genomic instability is particularly threatening due to long-term self-renewal and generation of abundant cellular descendants. Western blot and immunofluorescence revealed that H4K20me2 is dramatically decreased in *Kmt5b*-deficient MuSCs whereas H4K20me1 increased (Fig. 2a, Supplementary Fig. 2a). In contrast, H4K20me3 levels remained unchanged, probably due to direct conversion of H4K20me1 to H4K20me3 by KMT5C (Fig. 2a; Supplementary Fig. 2b). We did not detect obvious alterations of H4K20 mono-, di-, and tri-methylations during mitosis (Supplementary Fig. 2c), suggesting that H4K20me1-dependent kinetochore assembly and pericentric heterochromatin formation are not affected by loss of *Kmt5b*[22–24]. The intensity of H4K20me2 was mainly reduced in the euchromatin region and remained unchanged in DAPI-dense chromocenters of *Kmt5b^sKO^* MuSCs in S-phase (EdU+) (Supplementary Fig. 2a), indicating that aberrant mitosis in *Kmt5b^sKO^* MuSCs is not attributed to disrupted function of centromeres or pericentromeres. The reduction of H4K20me2 seems to be at odds with the strong accumulation of 53BP1+ foci in *Kmt5b^sKO^* MuSCs (Supplementary Fig. 1b–d), since H4K20me2 has been reported to recruit 53BP1 to DNA damage sites[25,26]. However, it is possible that the strong increase of H4K20me1 or compensatory activities of other histone methyltransferases such as KMT5C facilitate recruitment of 53BP1 in this condition. In WT (wildtype) MuSCs, genome-wide deposition of H4K20me1 showed a dynamic pattern during cell cycle progression with much lower levels during S-phase (EdU+) (Fig. 2b). In stark contrast, H4K20me1 increased predominantly in EdU+ MuSCs when *Kmt5b* was absent, although the protein levels of the H4K20me1 methyltransferase KMT5A decreased and the demethylase PHF8 increased only mildly, indicating that H4K20me1 accumulated in S-phase due to missing conversion of H4K20me1 to H4K20me2 mediated by KMT5B (Fig. 2b; Supplementary Fig. 2d).

To confirm that the catalytic activity of *Kmt5b* is crital for control of H4K20me1 levels in S-phase, we overexpressed human wildtype and catalytical-inactive KMT5B(hKMT5B S251A), carrying a point mutation in the catalytic domain[27], in Ctrl and *Kmt5b^sKO^* MuSCs (Supplementary Fig. 2e). Expression of mutant h*KMT5B* resulted in a dramatic loss of H4K20me2 but a strong accumulation of H4K20me1 in Ctrl MuSCs, mimicking the phenotype of *Kmt5b^sKO^* MuSCs (Supplementary Fig. 2f, g).

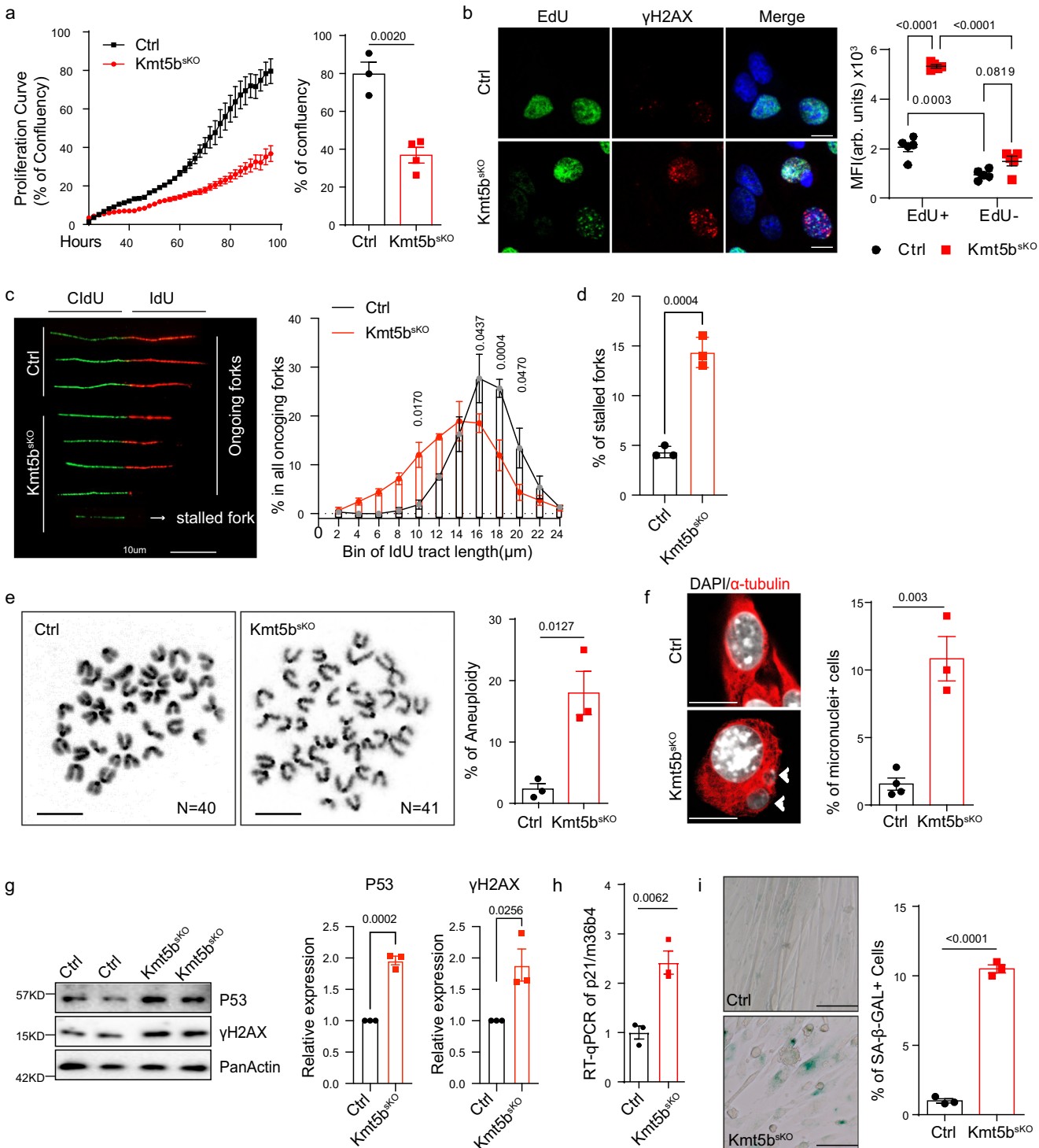

**Fig. 1 | *Kmt5b* prevents replication stress and genome instability in proliferating MuSC. a** Proliferation rate of *Ctrl* (*n* = 3 mice) and *Kmt5b^sKO^* (*n* = 4 mice) MuSCs quantified during time lapse imaging. Confluency of *Ctrl* and *Kmt5b^sKO^* MuSCs at the last time point is on the right. **b** Immunofluorescence staining for EdU and γH2AX in *Ctrl* and *Kmt5b^sKO^* MuSCs after 4 days in culture (*n* = 4 mice). DNA was stained by DAPI. Scale bar: 10 μm. Median of fluorescence intensity (MFI, arbitrary units (arb. units)) of γH2AX/EdU⁺ and γH2AX/EdU⁻ MuSCs are shown on the right. Data are shown as mean ± SEM; Two-way ANOVA. **c** Representative images of DNA fibers labeled with CldU and IdU. Scale bar: 10 μm. Quantification of fork progression in *Ctrl* and *Kmt5b^sKO^* MuSCs is shown on the right (*n* = 3 mice). Data are shown as mean ± SEM; Two-way. **d** Percentage of stalled folks in *Ctrl* and *Kmt5b^sKO^* MuSCs in all fibers (*n* = 3 mice). **e** Images of DAPI-stained metaphase chromosome spreadings prepared from *Ctrl* and *Kmt5b^sKO^* MuSCs (*n* = 3 mice). Scale bar: 10 μm.

Quantification of aneuploid MuSCs is on the right. **f** Immunofluorescence staining for α-tubulin/DAPI showing *Kmt5b^sKO^* MuSCs with micronuclei (indicated by arrowheads). Scale bar: 10 μm. Percentage of *Ctrl* (*n* = 3 mice) or *Kmt5b^sKO^* (*n* = 4 mice) MuSCs with micronuclei is on the right. **g** Western blot analysis of pP53 and γH2AX in *Ctrl* and *Kmt5b^sKO^* MuSCs after 4 days in culture. Pan-Actin was used as loading control. Band intensities were quantified, corresponding plots are shown on the right (*n* = 3 mice). **h** RT-qPCR analyses of *p21* expression in *Ctrl* and *Kmt5b^sKO^* MuSCs (*n* = 3 mice). *36b4* was used as reference gene. **i** SA-β-GAL staining of *Ctrl* and *Kmt5b^sKO^* MuSCs after 7-days in proliferation medium (*n* = 3 mice). Scale bar: 20 μm. Quantification of SA-β-GAL positive cells is on the right. Data are shown as mean ± SEM. Unpaired two-sided Student's *t*-tests were employed for all panels except for (**b**) and (**c**). Source data are provided in the Source Data file.

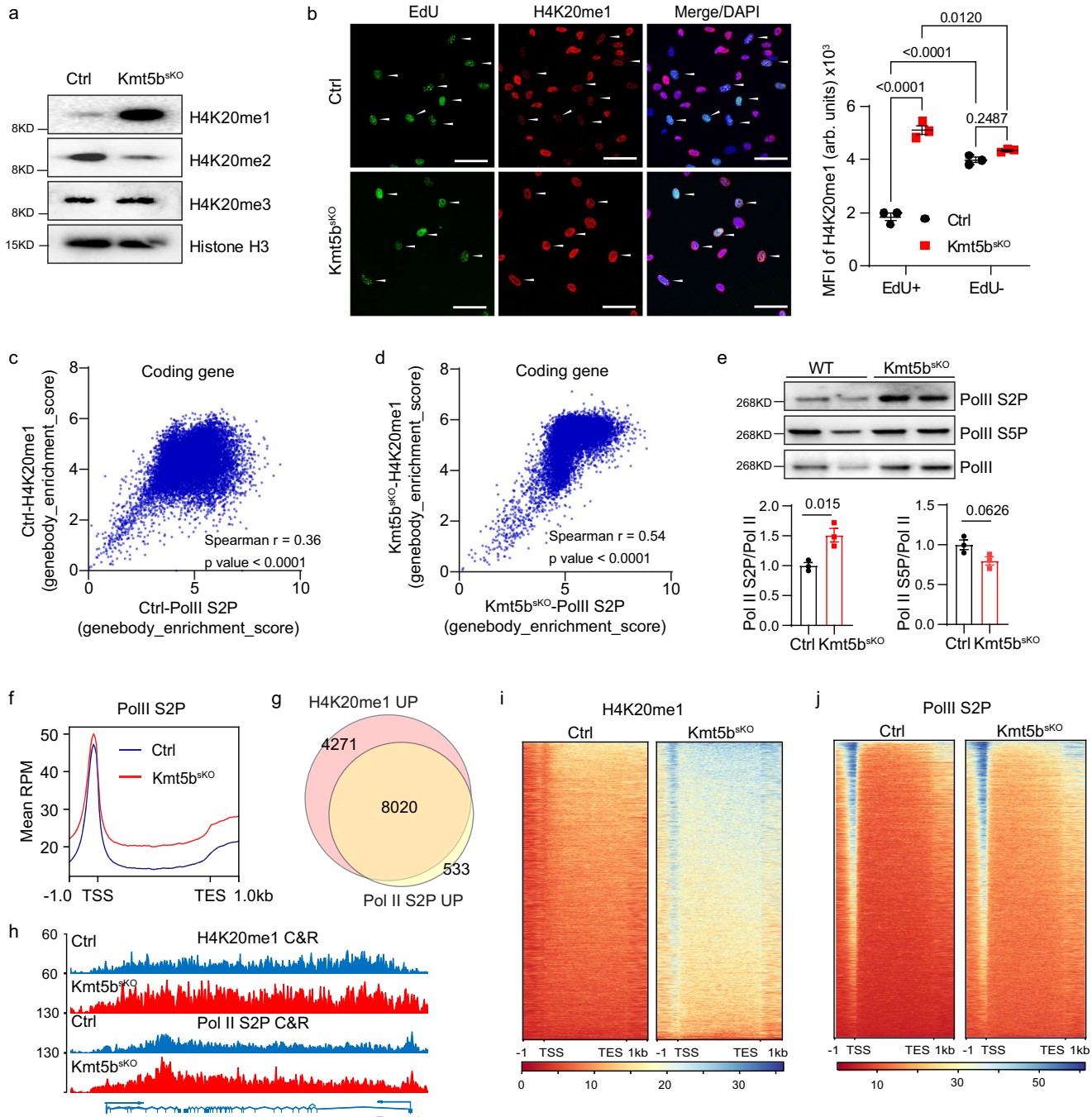

**Fig. 2 | Inactivation of *Kmt5b* increases accumulation of H4K20me1 and occupancy of Pol II S2P on genes in S-phase. a** Western blot analysis of histone H4 lysine 20 methylation in proliferating *Ctrl* and *Kmt5b^sKO* MuSCs. Histone H3 was used as loading control (*n* = 2 mice). **b** Immunofluorescence analysis of EdU incorporation and H4K20me1 enrichment in *Ctrl* and *Kmt5b^sKO* MuSCs after 3-days in culture. DNA was stained by DAPI. Cells in S-phase (EdU⁺) are indicated by arrowheads. Scale bar: 10 μm. Median of fluorescence intensity (MFI, arbitrary units (arb. units)) of H4K20me1 in EdU⁺/EdU⁻ MuSCs are shown on the right (*n* = 3 mice). Two-way ANOVA. **c, d** Scatter plot showing correlation of H4K20me1 enrichment and Pol II S2P binding within gene bodies of *Ctrl* (**c**) or *Kmt5b^sKO* (**d**) MuSCs. Spearman's correlation coefficient *r* values are indicated. Two-tailed *P*-values indicate the significance of the correlation determined by *r* values and sample size. **e** Western blot analysis of Pol II S2P and S5P in proliferating *Ctrl* and *Kmt5b^sKO*

MuSCs. Pol II was used as loading control. Band intensities were quantified and normalized values are in the lower panels (*n* = 3 mice). Unpaired two-sided Student's *t*-test. **f** Averaged profiles of intragenic Pol II S2P occupancy on all coding genes from −1 kb upstream of transcription start sites (TSS) to +1 kb downstream of transcription end sites (TES) in proliferating *Ctrl* and *Kmt5b^sKO* MuSCs. **g** Venn diagram of genes with increased accumulation of H4K20me1 (H4K20me1 UP) and Pol II S2P (Pol II S2P UP) occupancy within gene bodies of *Kmt5b^sKO* MuSCs compared to *Ctrl* MuSCs. **h** Genome browser tracks displaying H4K20me1 and Pol II S2P CUT&RUN signals in the *Rasa1* locus of *Ctrl* and *Kmt5b^sKO* MuSCs. *Y*-axis represents spike-in normalized read counts. **i, j** Heat maps representing H4K20me1 (**i**) and Pol II S2P (**j**) CUT&RUN signals from −1 kb upstream of TSS to +1 kb downstream of TES of overlapping genes from (**g**). Data are shown as mean ± SEM. Source data are provided in the Source Data file.

Accordingly, overexpression of WT but not the catalytically-inactive h*KMT5B* normalized H4K20me1 and H4K20me2 levels in *Kmt5b*[sKO] MuSCs during S-phase (Supplementary Fig. 2f, g). In line with these findings, DNA damage in *Kmt5b*[sKO] MuSCs during S-phase was markedly reduced by overexpression of WT but not the catalytically inactive h*KMT5B* (Supplementary Fig. 2h).

To explore whether accumulation of H4K20me1 in the euchromatin of *Kmt5b*[sKO] MuSCs during S-phase is associated with activation of transcription and elongation, we performed CUT&RUN experiments, thereby examining the genome-wide distribution of H4K20me1 and elongating RNA polymerase II phosphorylated at Ser2 (Pol II S2P)[28]. Consistent with previous data, enrichment of H4K20me1 over bodies of coding genes positively correlated with Pol II S2P occupancy in proliferating *Ctrl* and *Kmt5b*[sKO] MuSCs (Fig. 2c, d; Supplementary Fig. 2i). No positive correlation was found for typical non-coding genes including rDNA, snRNA genes and snoRNA genes, indicating that deposition of H4K20me1 is not relevant for RNA polymerase I and III-dependent transcription (Supplementary Fig. 2j, k). Genome browser views of mouse chromosome 9 illustrate that accumulation of H4K20me1 was specifically enhanced in actively transcribed regions of *Kmt5b*[sKO] MuSCs labelled by Pol II S2P binding (Supplementary Fig. 2l). We only detected elevation of Pol II S2P but not of Pol II S5P together with enhanced occupancy of Pol II S2P in gene bodies, demonstrating elevated transcriptional elongation in *Kmt5b*[sKO] MuSCs (Fig. 2e, f). In total, we identified 12,291 genes showing enhanced enrichment of H4K20me1 and 8553 genes with increased Pol II S2P occupancy in *Kmt5b*[sKO] MuSC. 8020 genes (93.8%) with enhanced Pol II S2P occupancy also showed an increase in H4K20me1 and were therefore considered as potential KMT5B targets (Fig. 2g–j).

## KMT5B represses transcription of a subset of coding genes and eRNAs during S-phase

To dissect the molecular mechanisms responsible for enhanced transcriptional activity in *Kmt5b*[sKO] MuSCs, we performed precision nuclear run-on transcription coupled with deep sequencing (PRO-seq) and determined proximal promoter pausing-release of Pol II in KMT5B target genes in control and *Kmt5b*[sKO] MuSCs. Profiling of 8020 potential KMT5B target genes revealed increased transcription activity at both TSSs and gene bodies (Supplementary Fig. 3a). Inactivation of *Kmt5b* did not change the pausing index (PI) of these genes, excluding the possibility that accumulation of H4K20me1 in *Kmt5b*[sKO] MuSCs facilitates release of paused Pol II in the proximity of TSSs as described for other cell types[29] (Supplementary Fig. 3b, c).

In addition to increased transcriptional activity of coding genes, we detected elevated expression of bi-directionally transcribed enhancer RNAs (eRNAs) from a large group of distal intergenic enhancer regions marked by H3K4me1 (802 out of 4414 enhancer regions, 18.2%) while only 42 eRNAs (0.9%) were transcriptionally downregulated in *Kmt5b*[sKO] MuSCs (Fig. 3a–c, Supplementary Fig. 3e). Increased eRNA transcription may indicate elevated enhancer activity[30], which was supported by a dramatic increase of active enhancer marks such as H3K27ac and H4K16ac in *Kmt5b*[sKO] MuSCs, concomitant with an increase of H4K20me1 levels. Accordingly the repressive histone mark H3K27me3 was downregulated (Fig. 3d; Supplementary Fig. 3e), suggesting that KMT5B suppresses eRNA transcription. Similar to H4K20me1, the increase of H3K27ac and H4K16ac occured predominantly in EdU+ *Kmt5b*[sKO] MuSCs (Supplementary Fig. 3f, g). KMT5B-affected enhancer regions showed enrichment of binding motifs for transcription factors such as KLF4, AP-1, MEF-2 and MYOD1, which might be required for increased eRNA transcription (Supplementary Fig. 3h). Taken together, the findings demonstrate that *Kmt5b* is indispensable for transcriptional silencing of a subset of coding genes and eRNAs, both of which are transcribed by Pol II.

To investigate whether enhanced transcriptional activity is mainly restricted to S-phase, we performed 5-Ethynyl-uridine (5-EU) incorporation assays in cultured MuSCs in combination with PCNA immunostaining. Importantly, EU signals remained unchanged in PCNA⁻ cells but were strongly enhanced in PCNA⁺ *Kmt5b*[sKO] MuSCs compared to control cells, suggesting that transcription activity is specifically increased during S-phase of *Kmt5b*[sKO] MuSCs (Fig. 3e). Overall, our results indicate that KMT5B-mediated conversion from H4K20me1 to H4K20me2 is essential for transcriptional silencing during S-phase.

## Loss of S-phase-specific transcriptional silencing leads to transcription-replication collisions and R-loop formation in MuSCs

Transcripts generated by aberrantly increased gene activity during S-phase may encounter replication forks, leading to TRCs and accumulation of uncontrolled R-loops. To demonstrate the occurence of TRCs, we performed proximity ligation assays using PCNA and Pol II S2P antibodies to detect co-localization of the replisome and transcription machinery in control and *Kmt5b*[sKO] MuSCs during S-phase (EdU⁺). The number of EdU⁺ MuSCs containing more than three PCNA-Pol II S2P PLA foci, the maximal value in single antibody-stained controls, was significantly increased after *Kmt5b* inactivation (Fig. 4a; Supplementary Fig. 4a). Notably, PCNA-Pol II S2P PLA foci in *Kmt5b*[sKO] MuSCs were mainly found within early replicating euchromatin but not within mid-late replicating chromatin (Fig. 4a, b). This finding suggests that TRCs occur primarily within actively transcribed euchromatin where TRCs remain unresolved until late S-phase. Of note, TRCs in S-phase *Kmt5b*[sKO] MuSCs were reduced to nearly control levels after expression of wildtype but not catalytic-inactive human KMT5B, demonstrating that collisions are due to failed conversion of H4K20me1 to H4K20me2 (Supplementary Figs. 4b, 2g). Since TRCs replication forks might stall and promote formation of genome-destabilizing R-loop[31,32], we next performed immunofluorescence staining using the S9.6 antibody to detect RNA:DNA hybrid (R-loop) on cryosections of injured TA muscle. The number of S9.6⁺/MyoD⁺ myoblasts was substantially increased in *Kmt5b*[sKO] muscle compared to control muscle (Supplementary Fig. 4c). Moreover, dot blots with genomic DNA revealed enhanced accumulation of RNase H-sensitive S9.6 signals in proliferating *Kmt5b*[sKO] MuSCs in vitro (Fig. 4c), indicating that loss of *Kmt5b* promotes formation of R-loops in MuSCs. Interestingly, increased formation of R-loops was found in S-phase rather than G1-phase of *Kmt5b*[sKO] MuSCs, suggesting dependency on TRCs (Supplementary Fig. 4d). We also detected increased levels of phosphorylated ATR and phosphorylated replication protein A (RPA32) at Ser33 by immuno-staining in *Kmt5b*[sKO] MuSCs, implying that TRCs and newly formed R-loops activate RPA32/ATR-P53 signaling[33] (Fig. 4d, e). Furthermore, increased levels of phosphorylated of RPA32 at Ser4/Ser8 and Thr21 indicated collapse of replication forks as a consequence of persistent TRCs in *Kmt5b*[sKO] MuSCs (Supplementary Fig. 4e, f).

To determine the genomic locations of R-loops and the correlation with transcription, DNA–RNA immunoprecipitation sequencing (DRIP-seq) was done using the s9.6 antibody, which yielded reproducible peaks within three biological replicates (Supplementary Fig. 4g)[34]. Consistent with dot-blot and immunofluorescence results, a large number of R-loop (8345 peaks) emerged in proliferating *Kmt5b*[sKO] MuSCs while only relatively few were lost (204 peaks) (Supplementary Fig. 4h). 54% of newly appearing R-loops were located within gene bodies of 2350 coding genes (>1 kb from TSS) and 34% were situated in intergenic regions (Fig. 4f; Supplementary Fig. 4i). 64.2% (1508 from 2350) of R-loop containing genes showed increased H4K20me1 and Pol II S2P enrichment, indicating that enhanced transcriptional activation is the main cause for increased R-loop formation in *Kmt5b*[sKO] MuSCs (Fig. 4g). R-loops were more frequent within potential KMT5B target genes >50 kb or showed a high GC content (Supplementary Fig. 4j, k). Of note, 40.3%

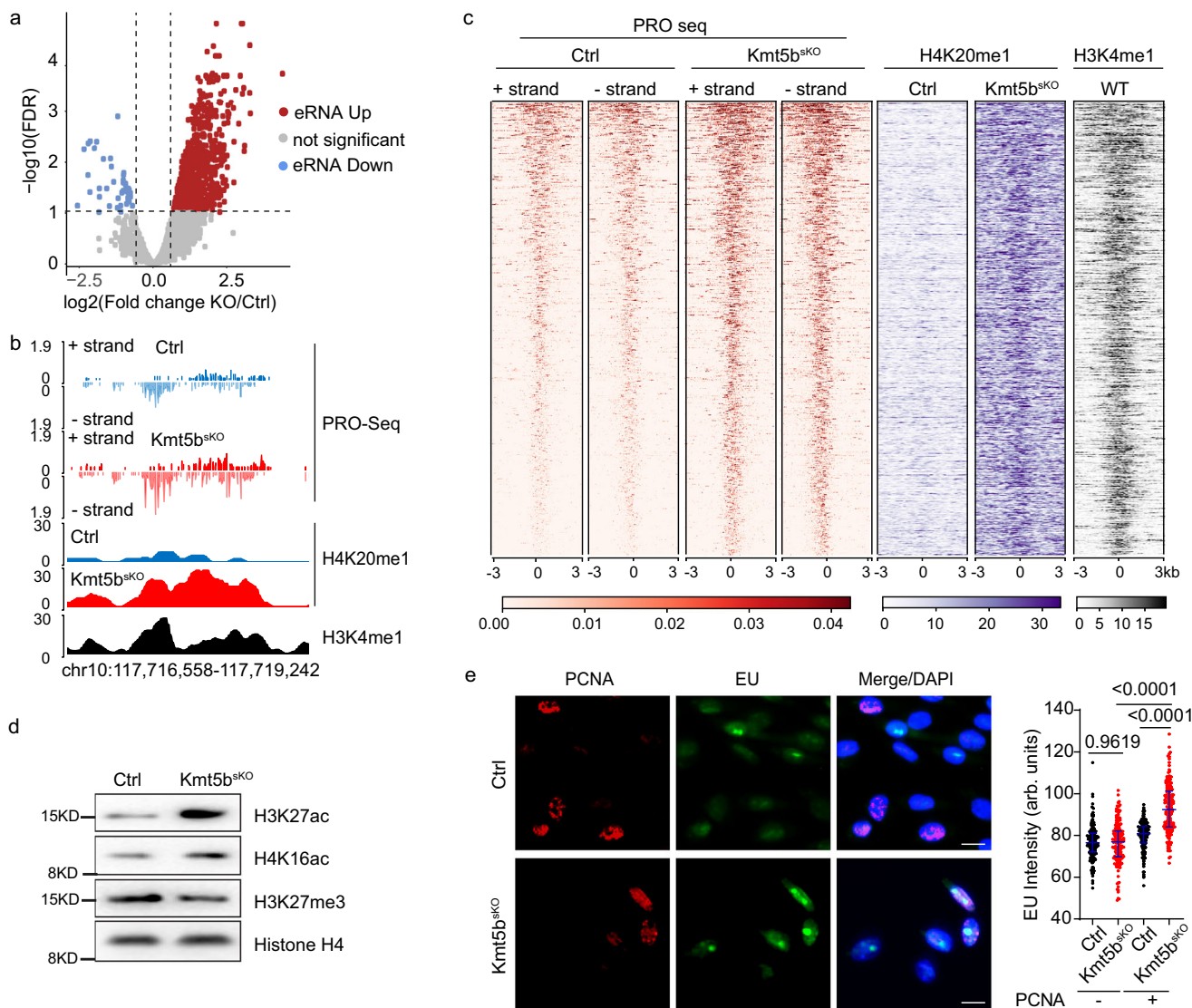

**Fig. 3 | Increased accumulation of H4K20me1 promotes transcription of a large group of coding genes and eRNAs. a** Volcano plot showing global changes of eRNA transcription in *Kmt5b^sKO* versus *Ctrl* MuSC. The log2-fold change of mutant/ *Ctrl* is represented on the *x*-axis. The *y*-axis shows the log10 of the false discovery rate (FDR). Dashed grey lines indicate significant cutoffs (FDR < 0.1 and log2 fold-changes >1.5). **b** Genome browser tracks of PRO-seq and H4K20me1 CUT&RUN signals across a representative enhancer region in *Ctrl* and *Kmt5b^sKO* MuSCs. H3K4me1 ChIP-seq (BioProject: PRJNA412267) identifies enhancer regions. Positive and negative values on the *y*-axis represent normalized library reads mapping to plus and minus strands, respectively. **c** Heatmaps representing the intensities of PRO-seq signals at enhancers identified by the NRSA v2 pipeline and corresponding H4K20me1 CUT&RUN signals in *Ctrl* and *Kmt5b^sKO* MuSC. H3K4me1 ChIP-seq data

from WT MuSCs (BioProject: PRJNA412267) are additional marker for enhancers. Rows are sorted by decreasing reads of eRNA. Color-scaled intensities are in units of RPM. **d** Western blot analysis of H3K27ac, H4K16ac and H3K27me3 in proliferating *Ctrl* and *Kmt5b^sKO* MuSCs. Histone H4 served as loading control (*n* = 2 mice). **e** Immunofluorescence staining for PCNA and EU incorporation in *Ctrl* and *Kmt5b^sKO* MuSCs after 3 days in culture. DNA was stained by DAPI. Scale bar: 10 μm. Quantification of EU incorporation intensities (arbitrary units (arb. units)) in PCNA⁻ (G1-phase) and PCNA⁺ (S-G2-phase) *Ctrl* and *Kmt5b^sKO* MuSCs is on the right. Error bars represent median ± interquartile range (*Ctrl* PCNA⁻, *n* = 225 cells; *Kmt5b^sKO* PCNA⁻, *n* = 180 cells; *Ctrl* PCNA⁺, *n* = 192 cells; *Kmt5b^sKO* PCNA⁺, *n* = 170 cells over 3 independent experiments). One-way ANOVA. Source data are provided in the Source Data file.

of newly forming intergenic R-loops were localized within enhancer regions, suggesting that enhanced eRNA transcription after *Kmt5b* inactivation contributes to R-loop formation (Fig. 4h, i). Altogether, our findings demonstrate that KMT5B-mediated conversion of H4K20me1 to H4K20me2 at genes bodies during S-phase prevents replication-transcription conflicts.

Replication stress and genome instability are tightly associated with the transition from normal to pre-cancerous or cancerous cells. Gene ontology and KEGG pathway analysis revealed that genes with newly acquired R-loop peaks are mainly associated with cancer development (Fig. 4j), of which several are frequently altered in human rhabdomyosarcomas, in particular genes related to DNA damage

response, cell migration and RAS signaling[20,35–37] (Fig. 4j; Supplementary Fig. 5a–d). Furthermore, 44% of early replicating fragile (ERFS) sites that were previously identified in mouse B cells to represent sites of active transcription associated with recurrent rearrangements during lymphoma formation showed increased R-loop formation (Supplementary Fig. 4l). In total, 127 genes with newly gained R-loop peaks are annotated as cancer drivers according to OncoKB, of which 25 genes are frequently mutated or undergo copy number alterations in human rhabdomyosarcomas. Such genes include oncogenes such as *Akt3*, *Braf*, *Cdk6*, *Hmga2*, *Yap1*, *Wwtr1*,*Rasa1*, *Agap2* and tumor suppressor genes *Nf1*, *Foxo1*, *Rb1*,*Arid4b* and *Robo1* (Fig. 4k, l; Supplementary Fig. 4m)[20,35,37,38].

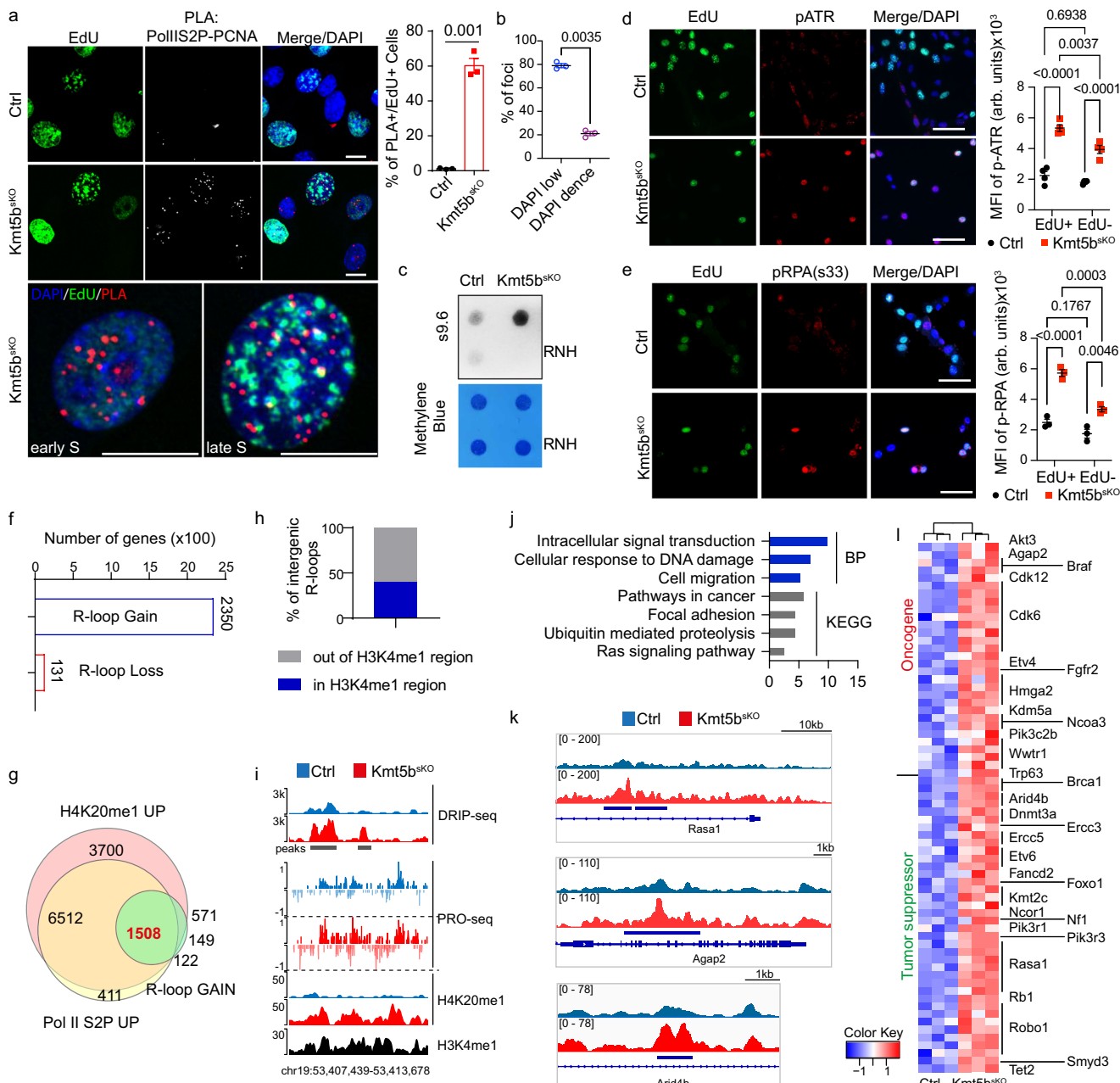

**Fig. 4 | Elevated transcriptional activity during S-phase increases transcription-replication collisions and enhances R-loop formation in *Kmt5b^{sKO}* MuSC.**
**a** Proximity ligation assay (PLA), visualizing co-localization of RNA PolII S2P and PCNA in EdU⁺ *Ctrl* and *Kmt5b^{sKO}* MuSCs (*n* = 3 mice). DNA was stained by DAPI. Representative images of *Kmt5b^{sKO}* MuSCs in early or late S phase are in the lower panel. Scale bar: 10 μm. Quantification of EdU⁺ cells with >3 PLA foci is on the right. **b** Quantification of PLA⁺ foci in DAPI low and dense regions in late S-phase (*n* = 3 mice). **a**, **b** Data represent mean ± SEM. Unpaired two-sided Student's *t*-test. **c** Dot-blot of R-loops using genomic DNA from *Ctrl* and *Kmt5b^{sKO}* MuSCs (*n* = 3 mice). RNase H (RNH) treated genomic DNA served as negative control and methylene blue staining as loading control. **d**, **e** Immunofluorescence analysis of (**d**) p-ATR/EdU and (**e**) p-RPA Ser33/EdU in *Ctrl* and *Kmt5b^{sKO}* MuSCs (*n* = 3 mice). Scale bar: 50 μm. MFI (arb. units) of (**d**) p-ATR and (**e**) p-RPA in EdU⁺/EdU⁻ MuSCs are shown on the right. Data are shown as mean ± SEM; Two-way ANOVA, Tukey correction. **f** Number of genes with gained and lost R-loops at genebodies after inactivation of *Kmt5b*. **g** Venn diagram of overlapping genes with increased H4K20me1 (H4K20me1 UP), increased Pol II S2P (Pol II S2P UP), and newly acquired R-loop peaks within genebodies of *Kmt5b^{sKO}* compared to *Ctrl* MuSCs. **h** Percentage of newly acquired intergenic R-loop near putative enhancer regions, marked by H3K4me1 enrichment. **i** Genome browser tracks of DRIP-seq, PRO-seq, and H4K20me1 CUT&RUN signals across a H3K4me1-enriched enhancer region of *Ctrl* and *Kmt5b^{sKO}* MuSCs. Positive and negative values on the *y*-axis represent normalized reads mapping to plus and minus strands, respectively. **j** Gene ontology analysis of biological processes (BP) and KEGG pathway analysis of genes showing newly acquired R-loop peaks within genebodies. **k** Genome browser tracks of DRIP-seq signals of *Rasa1*, *Agap2* and *Arid4b* genes in *Ctrl* and *Kmt5b^{sKO}* MuSC. *Y*-axis represents spiked-in normalized reads count. **l** Heatmap of newly acquired R-loop in genes promoting rhabdomyosarcoma formation. Source data are provided in the Source Data file.

## Repression of transcriptional elongation reduces TRCs and R-loop formation in *Kmt5b^{sKO}* MuSCs

To validate that enhanced transcriptional activity and particularly increased elongation during S-phase favors TRCs and R-loop formation in *Kmt5b^{sKO}* MuSC, we treated *Ctrl* and *Kmt5b^{sKO}* MuSCs with 10 μM 5,6-dichlorobenzimidazole 1-β-D-ribofuranoside (DRB), which specifically attenuates transcriptional elongation as indicated by reduced phosphorylation of Pol II at Ser2 in *Kmt5b^{sKO}* MuSCs (Supplementary Fig. 6a). Consistantly, *DRB* treatment reduced enhanced EU incorporation in PCNA⁺ *Kmt5b^{sKO}* MuSCs to the similar level as in *Ctrl* MuSCs

without significantly disturbing non-S-phase transcription (Fig. 5a). We assume that the selective effect of low-dose DRB treatment on transcription of PCNA+ *Kmt5b*ˢᴷᴼ MuSCs is caused by lowered chromatin compaction during S-phase. Attenuation of elongation decreased γH2AX levels in EdU+ mutant MuSC, indicating reduced DNA damage

(Fig. 5b; Supplementary Fig. 6a)[39,40]. Furthermore, DRB treatment normalized increased TRCs in EdU+ *Kmt5b*ˢᴷᴼ MuSCs to control levels (Fig. 5c), diminished aberrant R-loop accumulation detected by S9.6 antibodies (Fig. 5d), and ameliorated increased aneuploidy and micronuclei formation (Supplementary Fig. 6b, c). Importantly,

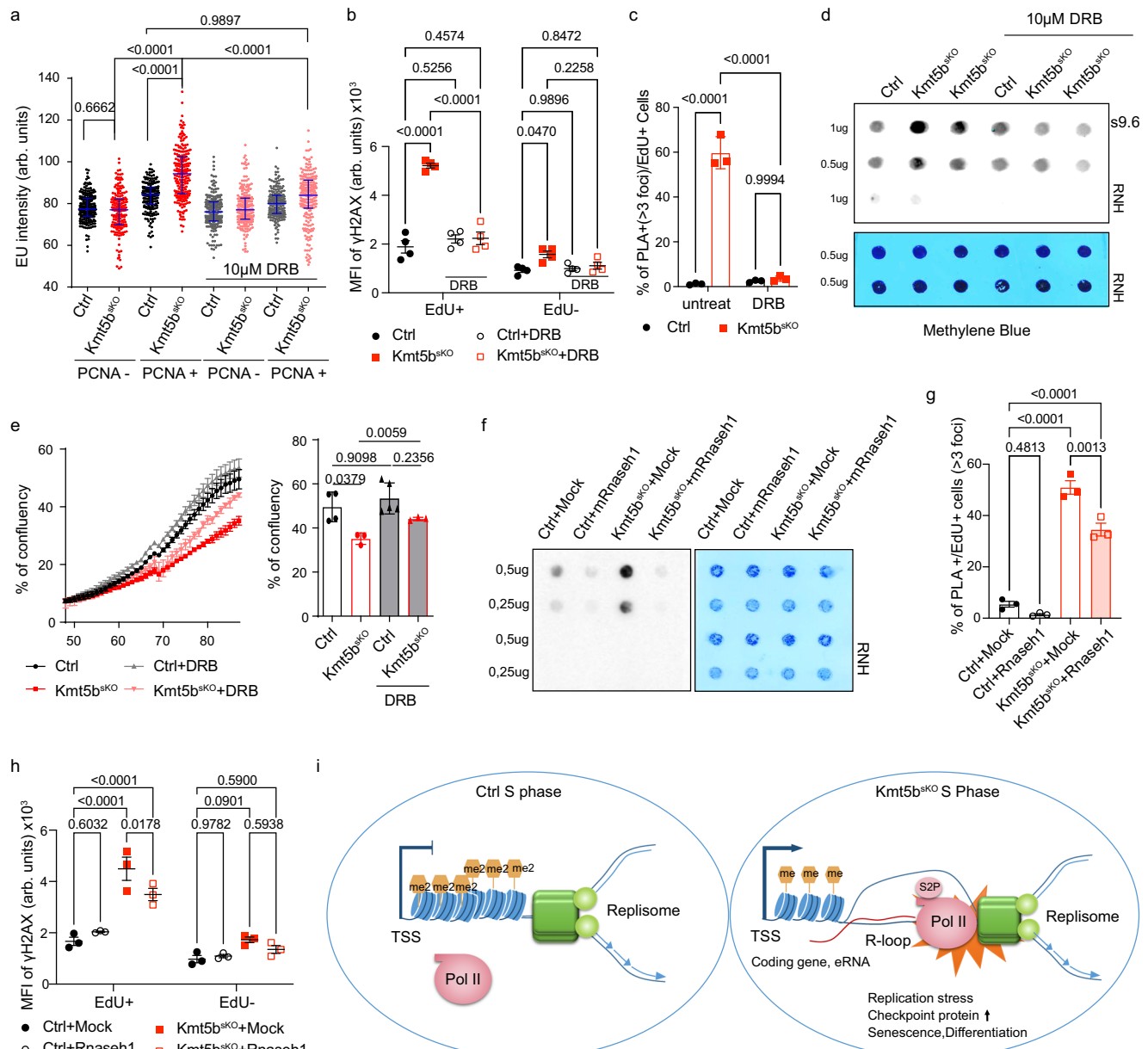

**Fig. 5 | Inhibition of S-phase transcription ameliorates TRCs and formation of R-loops. a** Quantification of EU labeling intensity (arbitrary units (arb. units)) in mock and DRB-treated *Ctrl* and *Kmt5b*ˢᴷᴼ MuSC, either in G1-phase (PCNA⁻) or S-G2-phase (PCNA⁺). Error bars represent median ± interquartile ranges (*Ctrl* PCNA⁻ n = 213 cells; *Kmt5b*ˢᴷᴼ PCNA⁻ n = 182 cells; *Ctrl* PCNA⁺, n = 203 cells; *Kmt5b*ˢᴷᴼ PCNA⁺, n = 212 cells; DRB + *Ctrl* PCNA⁻ n = 213 cells; DRB + *Kmt5b*ˢᴷᴼ PCNA⁻ n = 200 cells; DRB + *Ctrl* PCNA⁺, n = 204 cells; DRB + *Kmt5b*ˢᴷᴼ PCNA⁺, n = 222 cells, over 3 independent experiments). **b** Immunofluorescence analysis (median of fluorescence intensity, MFI (arbitrary units (arb. units)) of γH2AX/EdU⁺ and γH2AX/EdU⁻ in mock and DRB treated *Ctrl* and *Kmt5b*ˢᴷᴼ MuSCs (n = 4 mice). **c** Quantification of EdU⁺ cells with >3 PLA foci (PCNA:Pol II S2P) in mock or DRB-treated *Ctrl* and *Kmt5b*ˢᴷᴼ MuSCs (n = 3 mice). **d** Dot-blot analysis of R-loop formation in mock and DRB-treated *Ctrl* and *Kmt5b*ˢᴷᴼ MuSCs. Genomic DNA treated with RNase H served as negative control and methylene blue staining as loading control (n = 3 mice). **e** Proliferation curve of mock or DRB treated *Ctrl* and *Kmt5b*ˢᴷᴼ MuSCs assessed by time lapse

imaging. Confluency of each group at the last time point is shown in the right panel (*Ctrl* n = 4 mice; *Kmt5b*ˢᴷᴼ n = 3 mice; DRB + *Ctrl* n = 5 mice; DRB + *Kmt5b*ˢᴷᴼ n = 3 mice). **f** Dot blot analysis of R-loops by S9.6 antibody using genomic DNA after transduction of *Ctrl* and *Kmt5b*ˢᴷᴼ MuSCs with an empty vector (Mock) and a vector expressing *mRnaseh1*. Genomic DNA treated with RNase H served as negative control and methylene blue staining as loading control (n = 3 mice). **g** Quantification of EdU⁺ cells with >3 PLA foci (PCNA:Pol II S2P) in mock and mRnaseh1-overpressing *Ctrl* and *Kmt5b*ˢᴷᴼ MuSCs (n = 3 mice). **h** Immunofluorescence analysis (median of fluorescence intensity, MFI) of γH2AX/EdU⁺ and γH2AX/EdU⁻ in mock and *mRnaseh1*-overpressing *Ctrl* and *Kmt5b*ˢᴷᴼ MuSCs (n = 3 mice). **i** Simplified model of the role of KMT5B in coordinating transcription and replication in MuSC. Inactivation of *Kmt5b* in MuSCs leads to transcription-replication collision, subsequent genome instability, and cell cycle arrest. Data represent mean ± SEM. Two-way ANOVA. Source data are provided in the Source Data file.

reduction of TRC, R-loop formation and DNA damage after DRB treatment increased the proliferation rate of $Kmt5b^{sKO}$ MuSCs, probably due to reduction of aberrant mitoses (Fig. 5e; Supplementary Fig. 6d).

To investigate the causality between R-loop formation and transcription replication collisions, we overexpressed murine $Rnaseh1$ in $Ctrl$ and $Kmt5b^{sKO}$ MuSCs (Supplementary Fig. 6e). Interestingly, the percentage of PCNA-Pol II S2P PLA$^+$ cells as well as $\gamma$H2AX, p-RPA signals were only modestly reduced in EdU$^+$ $Kmt5b^{sKO}$ MuSCs, although R-loops were nearly eliminated in both $Ctrl$ and $Kmt5b^{sKO}$ MuSCs after expression of mRnaseh1 (Fig. 5f–h; Supplementary Fig. 6f). These findings suggests that R-loops only contribute to a minor degree to TRCs and that TRCs even without formation of R-loops are sufficient to induce genome damage. Taken together, the data indicate that increased transcription activity of a large group of coding genes and eRNAs during S-phase promotes collision of the replisome and transcriptional machinery, R-loop formation and genome instability of $Kmt5b^{sKO}$ MuSCs (Fig. 5i).

## Inhibition of the P53 response in $Kmt5b^{sKO}$ MuSCs abrogates senescence and promotes rhabdomyosarcoma formation

Loss of $Kmt5b$ provoked collision-associated R-loop formation on various cancer driver genes and activated pro-tumorigenic signaling in MuSCs, but no malignant transformation of $Kmt5b^{sKO}$ MuSCs occurred. We concluded that the marked upregulation of P53 and P21 in $Kmt5b^{sKO}$ MuSCs generated a barrier for neoplastic transformation by inducing cell cycle arrest and senescence. To prove this hypothesis, we generated $Pax7^{CreERT}/p53^{flox/flox}$ ($p53^{sKO}$) and $Pax7^{CreERT}/Kmt5b^{flox/flox}/p53^{flox/flox}$ ($DK$) mice, in which $p53$ expression is specifically deleted in MuSCs after tamoxifen injection (Fig. 6a). Depletion of $p53$ ameliorated the senescence-like phenotype of $Kmt5b^{sKO}$ MuSCs (Supplementary Fig. 7a) and significantly reduced expression of $p21$ (Fig. 6b). Furthermore, cell proliferation assays demonstrated that $DK$ MuSCs not only bypass cell cycle arrest but also proliferate more rapidly than $Ctrl$ and $p53^{sKO}$ MuSCs, leading to formation of aggregated, round clones (Fig. 6c; Supplementary Fig. 7b). Consistent with the in vitro results, we detected dramatically increased numbers of Ki67$^+$/PAX7$^+$ and EdU$^+$/PAX7$^+$ MuSCs in $DK$ compared to control, $p53^{sKO}$ and $Kmt5b^{sKO}$ muscles after EdU administration for 5 consecutive days under resting conditions (Fig. 6d; Supplementary Fig. 7c). Next, we challenged $DK$ muscles by induction of muscle regeneration. 10 days after CTX-induced muscle injury hyper-proliferative $DK$ MuSCs gave rise to massively increased numbers of unfused MYOG$^+$ myoblasts, which were mostly absent in control, $p53^{sKO}$ and $Kmt5b^{sKO}$ mice (Fig. 6e). Despite the massive accumulation of MYOG$^+$ myoblasts, we observed numerous new myofibers after one-time CTX injection, suggesting that $DK$ myoblasts do not encounter a major differentiation defect but rather undergo enhanced proliferation that outpaces differentiation (Fig. 6e).

Inactivation of $p53$ exacerbated genomic instability of $Kmt5b^{sKO}$ MuSCs as reflected by increased levels of $\gamma$H2AX, increased numbers of 53BP1 bodies-containing cells, exacerbated aberrant mitoses, aneuploidy, and micronuclei formation compared to $Kmt5b^{sKO}$ and $p53^{sKO}$ MuSC, confirming the critical function of $p53$ to eliminate potentially hazardous cells (Fig. 6f, g; Supplementary Fig. 7d–f). Depletion of $p53$ did not further enhance accumulation of aberrant R-loops in $Kmt5b^{sKO}$ MuSC, suggesting that accumulation of R-loop initiates a cascade of events, which cannot be prevented by P53, except for activation of senescence (Supplementary Fig. 7g). Importantly, all $DK$ mice rapidly developed tumors within or very close to the musculature of appendages or the trunk within 30 weeks after induction of $p53$ deletion (Fig. 6h, i). Histopathological analysis classified the emerging tumor as RMS, which stained positive for desmin, MYOD, and MYOG (Fig. 6j). Intriguingly, we identified copy number alterations (CNAs) of various oncogenes and tumor suppressors (e.g. $Yap1$, $Braf$, $Foxo1$, $Akt3$ and

$Rb1$) in RMS tumors developing in DK mice, which might play important roles for tumor formation and growth (Supplementary Fig. 7h). Genes with CNAs in tumors of DK mice also showed changes in R-loop formation in $Kmt5b^{sKO}$ MuSC. Furthermore, PLA$^+$ cells were still present on sections from 6 out of 10 mouse tumors isolated from DK mice, suggesting that TRCs persistent through tumor development (Supplementary Fig. 7i, j). Collectively, our findings demonstrate that loss of $p53$ in $Kmt5b^{sKO}$ MuSCs exacerbates genomic instability and prevents suppression of cell proliferation leading to unrestricted expansion of MuSC, eventually resulting in rhabdomyosarcoma formation (Supplementary Fig. 7k).

## Formation of human sarcomas is closely associated with reduced $Kmt5b$ expression

Oncogene induced replication stress and chromosome instability is a direct cause for CNAs, which are often found in human sarcomas but to a lesser extent in other malignancies[41–43]. To analyze whether $KMT5B$ plays a role in the formation of human sarcomas, we examined the TCGA pan-cancer data set and detected a significantly lower expression of $KMT5B$ in sarcomas than in most other malignancies (Supplementary Fig. 8a). We found that low expression of $KMT5B$ is associated with a shorter disease-free interval in all 29 cancer types analyzed, suggesting that $KMT5B$ plays a role in cancer progression and recurrence (Supplementary Fig. 8b). The low expression of KMT5B in sarcomas may be attributed to structural abnormalities of q13 on chromosome 11[35,44,45] as indicated by alterations of KMT5B copy numbers (Supplementary Fig. 8c–e). 70 out of 255 sarcoma samples (27.4%) showed a reduction of $KMT5B$ copy numbers (Supplementary Fig. 8f). Moreover, the promoter region of $KMT5B$ (cg1766288, cg2010049) is much more methylated in sarcomas compared to carcinomas such as THCA, PRAD, LUAD and BRCA, probably accounting for transcriptional repression of $KMT5B$ (Supplementary Fig. 8g–j). In conclusion, copy number loss combined with DNA hyper-methylation of promoter regions appears to contribute to inhibition of $KMT5B$ expression in human sarcomas. Of note, a large group of sarcomas showed low expression of both $KMT5B$ and $TP53$ (Supplementary Fig. 8c, j). Parallel analysis of TCGA PanCancer data uncovered that mutations of $TP53$ occur more frequent in tumors with low $KMT5B$ expression, supporting the idea that $Kmt5b$ and $p53$ collaborate to prevent tumor formation in mice (Supplementary Fig. 8k).

To validate the TCGA data, we monitored expression of $KMT5B$ and H4K20me1 levels in human cells, including a rhabdomyosarcoma cell line (RD), a osteosarcoma cell line (U2OS), two carcinoma cell lines (H1299, MCF7), two human myoblast cell lines and a human embryonic kidney cell line (HEK293). Consistent with the TCGA data, $KMT5B$ but not $KMT5A$ was dramatically lower in RD cells compared to the other cell lines, whereas H4K20me1 levels were substantially higher (Supplementary Fig. 9a–c). Moreover, we predominantly found copy number loss of $KMT5B$ in RD and U2OS cells and promoter hypermethylation accessed by HapII/MspII-qPCR in RD cells, which both seem to contribute to reduced expression of $KMT5B$ (Supplementary Fig. 9d, e). Decreased expression of $KMT5B$ in RD cells was associated with an increase in TRCs and stable R-loop formation, which is in line with the data from the mouse model, further confirming the results from TCGA demonstrating that low KMT5B is associated with high TRCs and activation of ATR signaling (Supplementary Fig. 9f, g). Interestingly, DRB treatment attenuated TRCs in a dose-dependent manner, but did not affect proliferation of RD cells, suggesting that TRCs are more important for malignant transformation than for proliferation of established cancer cell line, which may have been developed various means to bypass cell cycle checkpoints (Supplementary Fig. 9h, i).

## Discussion

Adult stem cells (ASCs) are not only required for organ maintenance and regeneration but are also a source for tumors[15]. Increased

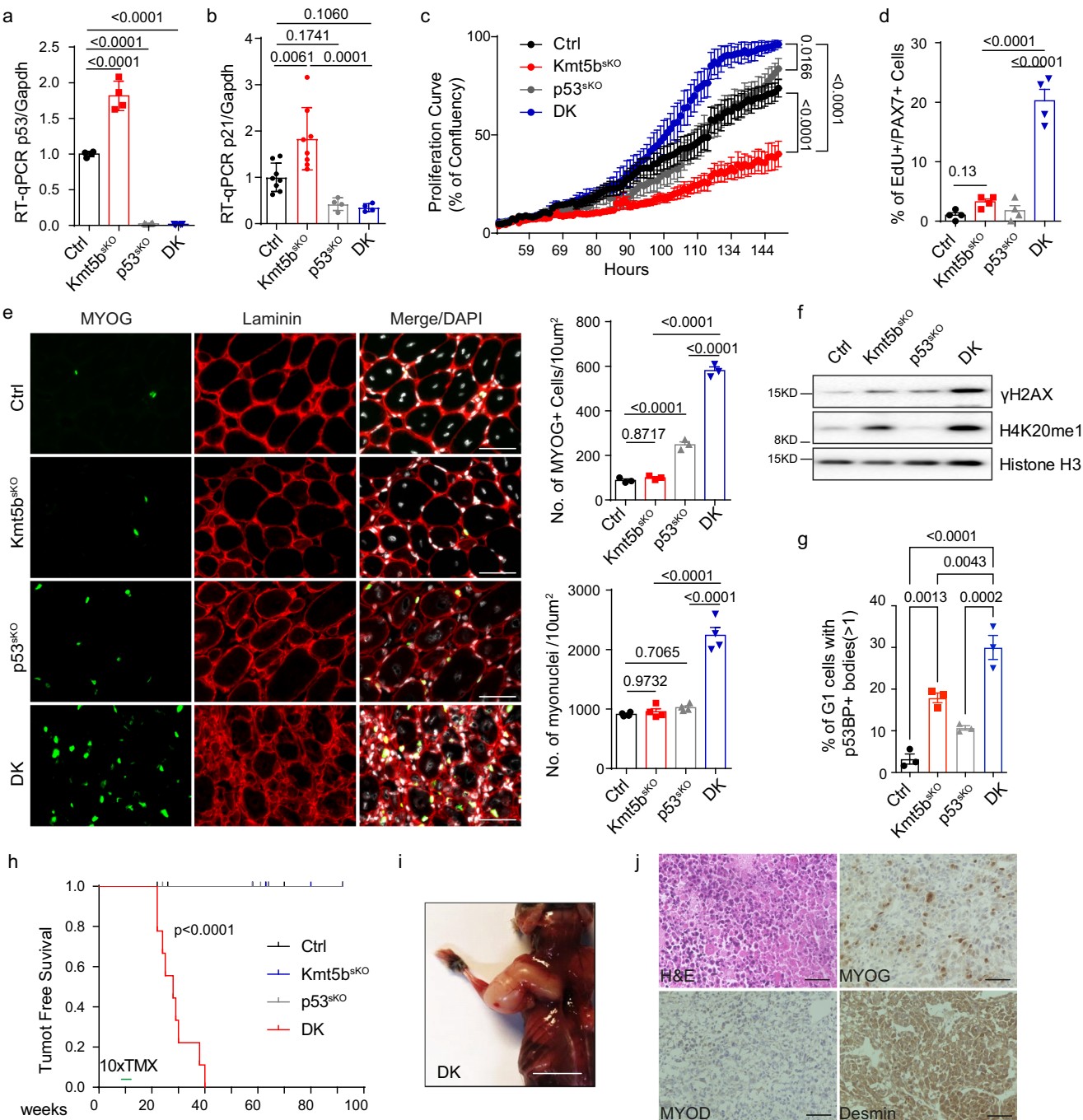

**Fig. 6 | Abrogation of cellular senescence and cell cycle arrest by inactivation of** ***p53*** **in** ***Kmt5b^sKO*** **MuSCs causes rapid formation of rhabdomyosarcomas. a, b** RT-qPCR analysis of *p53* (**a**) and *p21* (**b**) expression in *Ctrl*, *Kmt5b^sKO*, *p53^sKO* and *DK* MuSCs (*n* = 3 mice). *Gapdh* served as reference gene. **c** Proliferation of *Ctrl*, *Kmt5b^sKO*, *p53^sKO* and *DK* MuSCs assessed by time lapse imaging (*n* = 3 mice). Statistical significance at the last time point was calculated by One-way ANOVA. **d** Quantification of immunofluorescence analysis showing the presence of EdU⁺/PAX7⁺ MuSCs in TA muscles from *Ctrl*, *Kmt5b^sKO*, *p53^sKO* and *DK* littermates after 5 days of EdU administration (*n* = 4). **e** Immunofluorescence staining for MYOG and laminin on TA muscle cryosections from different mutant mice 10 days after CTX injection. Scale bar: 50 μm. Numbers of MYOG⁺ cells (upper right panel, *n* = 3 mice, each group) and myonuclei (lower right panel, *n* = 4 mice for each

group) per 10 microscopic fields were quantified. **f** Western blot analysis of H4K20me1 and γH2AX in *Ctrl*, *Kmt5b^sKO*, *p53^sKO* and *DK* MuSCs after 4 days of culture. Histone H3 served as loading control (*n* = 2 mice). **g** Quantification of *Ctrl*, *Kmt5b^sKO*, *p53^sKO* or *DK* MuSCs in G1-phase carrying more than one 53BP1 nuclear body after 4 days in culture (*n* = 3 mice). **h** Tumor-free survival Kaplan–Meier curves (logrank test) of *Ctrl* (*n* = 8), *Kmt5b^sKO* (*n* = 8), *p53^sKO* (*n* = 6) and *DK* (*n* = 10) littermates after tamoxifen injection. **i** Representative image of a *DK* mouse with a rhabdomyosarcoma in the forelimb after tamoxifen injection. Scale bar: 1 cm. **j** Haematoxylin and eosin (H&E) and immunostaining for MYOD, MYOG, and desmin on tumor cryosections from *DK* mice (*n* = 18 tumor samples). Scale bar: 50 μm. Data are shown as mean ± SEM. Statistical analysis in this Figure except (**h**) was performed by One-way ANOVA. Source data are provided in the Source Data file.

replication stress in ASCs, which occurs upon loss of quiescence during continuous regeneration, activation of oncogenes, or other reasons, causes genomic instability and favors tumor formation[1,46]. Here, we discovered that *Kmt5b* guards genomic integrity by suppressing transcription in S-phase, thereby preventing TRCs and replication stress, eventually protecting MuSCs from malignant transformation. We found that loss of *Kmt5b* led to accumulation of H4K20me1 prodominantly in S-phase, indicating that KMT5B-mediated conversion of H4K20me1 to H4K20me2 is a critical driver for reducing H4K20me1 before DNA synthesis. We did not detect significant changes of H4K20me3 in *Kmt5b*[sKO] MuSC, which is in line with previous reports describing a reduction of H4K20me3 in *Suw4-20h2/Kmt5c* but not in *Kmt5b* knockout mouse embryonic fibroblasts[25]. Since H4K20me3 but not H4K20me1/2 marks constitutive heterochromatin, we conclude that the genomic instability of *Kmt5b*[sKO] MuSCs is not caused by loss of constitutive heterochromatin. KMT5B/C-catalyzed formation of H4K20me2 is involved in DNA double strain break response to protect genome stability via 53BP1 binding[26], but recruitment of 53BP1 was not substantially reduced in *Kmt5b* deficient MuSC.

Accumulation of H4K20me1 has grave consequences for cellular physiology, mainly due to its critical roles in regulating transcription and DNA replication[47–49]. H4K20me1 has been proposed to increase transcriptional elongation by accelerating release of paused Pol II[48]. Surprisingly, we did not detect alterations of the Pol II pausing index in *Kmt5b*[sKO] MuSC, despite enhanced intragenic accumulation of H4K20me1 and Pol II Ser2 in a large group of genes, arguing for a more complex role of H4K20me1 and KMT5B in the regulation of transcription. In addition, we detected a massive increase of eRNA transcription in MuSCs after loss of *Kmt5b*. Regions generating eRNAs often contained binding motifs for the pluripotency factor KLF4, which binds to genomic regions generating G0/G1-specific eRNAs[50]. Inactivation of *Kmt5b* led to the specific increase of eRNAs that are normally expressed only at low levels during S-phase, supporting the hypothesis that KMT5B suppresses RNA Pol II-dependent transcription primarily in S-phase. We reason that aberrantly increased transcription of eRNAs, synergizes with the increase of H4K20me1 on gene bodies to increase transcriptional activity during S-phase. Nonetheless, further research is necessary to identify eRNA target genes and to rule out the possibility that increased eRNA production is merely an epiphenomenon of enhanced S-phase transcription.

Replication stress and TRC formation are enhanced by elevated expression of oncogenes such as *Ras*, *Mos*, *Myc*, *Cdc25A* or Cyclin E, which enhance replisome activity but also stimulate transcription[4]. Thus, either an increase of transcription or replication may create an imbalance, resulting in TRCs. We assume that increased transcriptional activity but not enhanced replication is the main reason for TRCs in Kmt5b[sKO] MuSC, because of the following reasons: (i) *Kmt5b*-deficent MuSCs show increased transcription specifically during S-phase. (ii) KMT5B/C mediated H4K20 methylation enhances formation of origins of replication and facilitates recruitment of the ORC1 complex[51]. Accordingly, loss of *Kmt5b* might reduce rather than increase replisome activity, often seen in cells overexpressing certain oncogenes[4]. We did not investigate whether increase of H4K20me1 and loss of *Kmt5b* alter the number or activity of origins of replication, which leaves some uncertainties. The critical role of enhanced transcription in S-phase for TRCs in *Kmt5b*[sKO] MuSCs is also supported by the efficient prevention of TRC and R-loop formation via treatment with low doses of DRB, which attenuates transcriptional elongation. Treatment with DRB also released the cell cycle block, and prevented DNA damage and genome instability in *Kmt5b*[sKO] MuSC.

Increased S-phase transcription can result in two different forms of TRCs with either co-directional or head-on orientations. Several studies reported that collisions between RNAPs and the replication machinery resulting from head-on orientation are stable, promoting R-loop formation and activation of the ATR pathway. In contrast, co-transcriptional R-loops are assumed to be relatively transient[31,32]. We found that collision foci in *Kmt5b*[sKO] MuSCs formed in early replicating regions and remained unresolved during late S-phase, suggesting that the observed TRCs resulted from head-on collisions. However, definitive proof for this hypothesis is missing.

Does accumulation of R-loops happens because of enhanced transcription in *Kmt5b*[sKO] MuSC, without requiring TRCs? Previous studies indicate that global stimulation of transcription by oncogenes such as HRAS[V12] is able to cause replication stress due to induction of R-loop accumulation[52]. In contrast, we are facing a different scenario in *Kmt5b*[sKO] MuSCs, in which elevated S-phase transcription causes TRCs and subsequent accumulation of unresolved R-loops that are less relevant for genomic instability than TRCs. This conclusion is based on several lines of experimental evidence: (i) only 1508 genes from 8020 transcriptional upregulated genes showed R-loop formation at gene bodies, suggesting that transcription activational alone is not sufficient to cause formation of stable R-loops; (ii) using purified genomic DNA from FACS-isolated S- and G1-phase MuSCs, we demonstrate that R-loops mainly accumulate during S-phase rather than in G1-phase, indicating that increased R-loops are associated with DNA replication; (iii) genomic mapping of R-loops by DRIP-seq revealed strong accumulation of aberrant R-loops mainly in gene bodies but not in TSS and TES, where R-loops involved in transcriptional regulation normally localize to exert their function for transcription pausing and termination; (iv) elimination of R-loops by RNaseH1 overexpression attenuated TRCs and DNA damage only to a very limited extent, clearly indicating that TRCs do not necessarily require the presence of R-loops[53]. Therefore, we conclude that loss of *Kmt5b* induces TRCs, unresolvable R-loops and collapse of the replication fork, which in combination causes DNA under-replication (UR-DNA), resulting in aberrant chromosome segregation and subsequent DNA lesions during mitosis. This conclusion is also supported by the accumulation of UR-DNA, marked by increased numbers of T53BP1[+] nuclear bodies in the G1-phase of *Kmt5b*[sKO] MuSC. Depletion of *p53* did not alter the numbers of aberrant R-loops in *Kmt5b*[sKO] MuSC, demonstrating that accumulation of aberrant R-loop activates p53-signaling and not vice versa.

Soft tissue sarcomas with low mutational loads including rhabdomyosarcoma are characterized by complex chromosomal aberrations such as copy-number alteration (CNA)[41,54]. Fragile CNA hotpots usually correspond to large and active transcription units in both human and mouse cells[55]. In line with these findings, we observed that TRC-induced R-loops in *Kmt5b*[sKO] MuSCs preferentially form in large genes and often overlap with fragile early replication sites[56]. Such fragile early replication sites are frequently found in oncogenes and tumor suppressors, and contribute to CNAs, favoring tumorigenesis. Accordingly, tumors in DK mice carried CNAs in genes that develop R-loops after *Kmt5b* depletion. CNA-containing genes included oncogenes showing copy number gain such as *Yap1*, *Braf*, *Foxo1*, and *Akt3* and tumor suppressors with copy number loss such as *Rb1*, which may favor tumor growth.

TRCs in *Kmt5b*-deficient cells may persistently occur throughout tumor development but may not be required for tumor growth. Interestingly, 6 out of 10 tumors from different DK mice showed cells with TRCs, which were completely absent in the other four tumors. Tumor cells may establish efficient ways for resolving TRCs to evolve into rapidly proliferating cells, such as via RNAP skipping, repriming, eviction and degradation, fork cleavage and re-ligation.

Analysis of TCGA data revealed that human sarcomas are characterized by reduced expression levels of *KMT5B* due to copy number loss and DNA hyper-methylation. In agreement with the findings in *DK* mice, we observed a high coincidence of alterations in human *TP53* and *KMT5B/KMT5B* genes in pan-cancer samples, which were not restricted to rhabdomyosarcomas but also present in different sarcoma types. This observation suggests that *KMT5B* functions as a general tumor suppressor in human sarcomas. Experiments using human

rhabdomyosaroma (RD cells) and cancer cell lines, validated these results. Human RD cells line showed low expression of *KMT5B* with high levels of TRCs and ATR activation. Surprisingly, proliferation of RD cells was not affected by treatment with DRB, despite lowering of TRCs, questioning the therapeutic potential of DRB for the treatment of KMT5B-low tumors.

Taken together, we provide proof of principle that an epigenetic modifier, repressing S-phase transcription by converting H4K20me1 to H4K20me2, is instrumental to prevent sarcoma formation. The lower disease-free interval of sarcoma patients with reduced expression of KMT5B compared to patients with higher KMT5B levels suggest that KMT5B level might be a usefull prognostic marker.

## Methods

### Animals

The *Kmt5b*$^{flox/flox}$ mouse strain was generated by the European Conditional Mouse Mutagenesis Program (EUCOMM) and was described previously[11]. The *p53*$^{flox/flox}$ mouse strain was obtained from the Jackson Laboratory (6.129P2-Trp53tm1Brn/J) and was described previously[57]. Generation of *Pax7*$^{CE}$ mice has been published[58]. All mice used in this study were healthy and not treated with drugs prior to the studies and were not used in previous procedures. None of the determined parameters in this study correlated with animal sex. All mouse strains were backcrossed and maintained on a C57BL/6 genetic background. Primers used for genotyping of individual transgenes are listed in Supplementary Data 1. Tamoxifen (Sigma) was administered to 8–12 week-old mice intraperitoneally at 0.05 mg/g of body weight per injection. Cardiotoxin (0.06 mg/ml, Sigma) was injected into tibialis anterior muscles in a volume of 50 μl. EdU incorporation was assayed in 13–15 weeks-old mice after injection of EdU for consecutive 5 days using a final concentration of 5 mg/kg body weight[59]. All animal experiments were done in accordance with the Guide for the Care and Use of Laboratory Animals published by the US National Institutes of Health (NIH Publication No. 85-23, revised 1996) and were reviewed and approved by Animal Rights Protection Committee of the State of Hessen (Regierungspraesidium Darmstadt, Wilhelminenstr. 1–3, 64283 Darmstadt, Germany) with the project number B2/1137.

### MuSC purification and culture

MuSC isolation and purification were performed according to established methods[60]. Briefly, limb and trunk muscles were minced, digested with 100 CU Dispase (BD) and 0.2% type II collagenase (Worthington Biochemicals) for one hour, and consecutively filtered through 100 μm, 70 μm, and 40 μm cell strainers (BD). Cells were precipitated and separated on a discontinuous Percoll gradient consisting of 70% Percoll overlaid with 30% v/v Percoll. Mononuclear cells were collected at the 70/30 interphase and subjected to MACS satellite cell isolation kit (Miltenyi Biotec.). MACS purified MuSCs were cultured on Matrigel-coated 96-well μClear plates (BD Biosciences, Greiner) in DMEM medium with 20% FCS, LIF (10 ng/ml) and bFGF (5 ng/ml)[60,61]. DRB treatment was performed two days after seeding for 48 h.

### Immunofluorescence and histology assays

Cultured cells were fixed in 4% paraformaldehyde or cold methanol optimized for the primary antibodies. Frozen muscle sections (5–10 μm) were fixed in cold acetone for 10 min. The primary antibodies used for immunohistochemical staining are listed as follows: Anti-γH2AX (Cell Signaling #2577, 1:1000), Anti-53BP1 (Abcam #ab36823, 1:1000), Anti-α-tubulin (clone B-5-1-2) (Sigma–Aldrich #T6074,1:2000), Anti-centrin-2 (Merck Millipore #ABE480, 1:1000), Anti-Ki67 (Abcam #ab15580, 1:1000), Anti-H4K20me1 (Abcam #ab9051, 1:2000), Anti-H4K20me2 (Diagenode #C15200205, 1:2000), Anti-H4K20me3 (Abcam #ab9053, 1:2000), Anti-phospho-RPA32 (Ser33) (Bethyl #A300-246A-M, 1:1000), Anti-phospho-RPA32 (S4/S8) (Bethyl #A300-245A-M, 1:1000), Anti-phospho-ATR (Cell Signaling

#2853, 1:1000), Anti-laminin (Abcam #ab11575, 1:2000), Anti-MYOD1 (Abcam #ab64159, 1:1000), Anti-myogenin (F5D, BD Bioscience #556358, 1:1000), Anti-PAX7 (R&D System #MAB1675, 1:1000), Anti-desmin (Sigma Aldrich #D8281, 1:1000). TUNEL assays to monitor apoptosis were carried out with the In Situ Cell Death Detection Kit (Roche) according to the manufacturer's protocol. Proximity ligation assay was performed using the Duolink In Situ kit (Sigma–Aldrich DUO92102) with rabbit anti-Pol II S2P (Abcam #ab5095, 1:1000) and mouse anti-PCNA(PC10, Abcam # ab29, 1:1000) antibodies. For controls, the PLA assay was performed with only a single antibody, which resulted in not more than 3 foci per cell. Therefore, only cell containing more than 3 foci was counted as positive. Senescent cells were detected based on β-galactosidase activity by the Senescence Cells Histochemical Staining Kit (Sigma). Haematoxylin and eosin staining was used to visualize muscle fibers. The samples were imaged on confocal (Leica), Axioimager (Zeiss) microscopes or Kenyence microscope. Median fluorescence intensity of immnunofluorescent staining in DAPI$^+$ regions was quantified by FIJI.

### DNA fiber assay

Cells were pulse labelled with 25 mM CldU and 250 mM IdU for 20 min and harvested . DNA fibre spreads were prepared by spotting 4 ul of cells ($1.75 \times 10^5$ cells per ml in PBS) onto superfrost slides followed by gentle mixing with 7 ul lysis buffer (0.5% SDS, 200 mM Tris-HCl pH 7.4 and 50 mM EDTA). The slide was kept horizontally for 8 min at RT and tilted (15°–30°) to allow the drop to run slowly down the slide. DNA spreads were air dried and fixed in methanol/acetic acid (3:1) at RT for 40 min. After denaturation in 2.5 M HCl for 1.5 h at RT, fibre spreads were blocked in 5% BSA/PBS for 1 h and incubated with rat anti-bromodeoxyuridine detecting CldU (BU1/75, Acris 1:1,000), and mouse anti-bromodeoxyuridine detecting IdU (B44, Becton Dickinson 1:500) at 4 °C overnight. Slides were washed three times with PBST and incubated with anti-rat IgG Alexa Fluor 594 (Invitrogen #A11007, 1:500) and anti-mouse IgG Alexa Fluor 488 (Invitrogen #A11001, 1:500) for 1.5 h. The wash was repeated three times and slides was mounted with 30 μl moviol. Images were acquired using an Axioimager (Zeiss) upright microscope with Zen2 6.1.7601 software and analysed using FIJI (https://imagej.net/software/fiji/). At least 100 fibres were measured for each condition in each independent experiment.

### Measurment of cell cycle phases by FACS analysis

Cells were pulse-labeled with 10 μM EdU for 30 min before fixation. EdU incorporation was visualized using the Click-iT™ EdU Alexa Fluor™ 488 Flow Cytometry Assay (Invitrogen), according to the manufacturer's protocol, followed by DAPI staining on ice for more than 1 h. FACS analysis was perform using the BD LSRFortessa™ Cell Analyzer and data were analyzed using FlowJo software. Fixed S-phase and G1-phase cells were sorted using the BD FACSAria™ III Cell Sorter and FACSDiva™ v8 Software. The detailed gating strategy to identify cells in G0/G1 and S,G2/M phases is described in Supplementary Fig. 10.

### Plasmids construction and retroviral transduction of MuSCs

Retroviral plasmids (pMX-empty, pMX-HA-hKMT5BWT-V5, pMX-mRnash1-HA) were constructed by cloning hKMT5B and mRNaseh1 cDNA PCR products into the pMX vector. The point mutation (pMX-HA-hKMT5BMut (S251A)-V5) was generated using the Quickchange II Site-Directed Mutagenesis Kit (Agilent). Primers for cloning and mutagenesis are listed in Supplementary Data 1. Plasmids were transfected using PEI to Platinum-E (Plat-E) cells, which were ~70% confluent in 10 cm dishes. The transfection medium were replaced with fresh DMEM medium 12–14 h post transfection. The supernatant containing retrovirus was collected 36 h post transfection and filtered through a 0.22 um filter. Next, 10% FCS, bFGF (5 ng/ml) and polybrene (8 μg/mL) was added to the viral supernatant before administration to MuSCs, which were seeded 24 h in advance. 24 h after transduction, the

medium was replaced with MuSC growth medium. Cells were harvested 2–3 days later for immunostaining and DNA extraction after reaching 70% confluency.

## DNA methylation-sensitive PCR

Cell pellets were digested over night at 50 °C with digestion buffer (100 mM NaCl, 10 mM Tris-HCl, pH 8, 25 mM EDTA pH 8, 0.5% Sodium dodecyl sulfate, and freshly added 0.1 mg/ml proteinase K). DNA was isolated by phenol/chloroform/isoamyl extraction. 1 μg purified DNA was subjected to enzymatic digestion with the methylation-sensitive enzymes HpaII and MspI (NEB), followed by qPCR analysis with primers flanking the restriction site (Supplementary Data 1). DNA input without digestion was used as control.

## EdU and EU incorporation assays

Cells were pulse labeled with 10 μM EdU for 10 min before fixation and EdU incorporation was visualized using the Click-iT EdU kit (Invitrogen) according to the manufacturer's protocol. To detect nascent RNA, proliferating MuSCs were incubated with 5-Ethynyl uridine (EU) (100 μM) for 1 h and EU incorporation in nascent RNA was visualized by Click-iT EU imaging kit (Invitrogen). Time-lapse imaging and analysis were performed using an Incucyte Live-Cell Imaging System and software (Essen Instruments).

## Metaphase spreads

MuSCs after 4 days culture in vitro were incubated with 50 μl of colcemid (final concentration of 100 ng/ml) at 37 °C for 1 h. Cells were then trypsinized, pelleted by centrifuge, and re-suspended in 500 μl of culture medium. 15 ml pre-warmed 75 mM KCl was added drop-wise into the cell suspension. After incubation at 37 °C for 15 min, the cells were spin down again and re-suspended in 500 μl of remaining supernatant. 15 ml ice-cold fixative solution (methanol/acetic acid 3:1) was added drop-wise into the cells. The fixation was repeated for 3 times. After the last spin down, 35 μl of the re-suspended MuSCs were dropped onto a slide and dried in the humidity chamber for 30 min. The chromosomes were stained with DAPI an d imaged on confocal (Leica) microscope (Las X 3.5.7.23225) with high magnifence and deconvolution. For each biological sample, 30 metaphase nuclei were analyzed. The chromosome number was quantified by ImageJ.

## Western blot

In vitro cultured satellite cells were harvested, washed with ice cold PBS and lysed in cell lysis buffer (20 mM Tris pH 7.5, 400 mM NaCl, 1 mM EDTA, 1 mM EGTA, 1% Triton X-100, 2.5 mM Sodium pyrophosphate, 1 mM β-glycerophosphate, 1 mM $Na_3VO_4$, 1 μg/ml Leupeptin). 10 μg of whole cell lysates were subjected to SDS-PAGE. Primary and secondary antibodies were: Anti-γH2AX (Cell Signaling #2577, 1:1000), Anti-P53 [PAb 240] (Abcam #ab26, 1:1000), Anti-pan-actin (Cell Signaling #4968, 1:1000), Anti-histone H3 (Abcam #ab18521, 1:2000), Anti-V5 Tag (ThermoFisher #R960-25, 1:1000), Anti-HA tag (Abcam #ab9110, 1:1000), Anti-SETD8 (Millipore #06-1304, 1:1000), Anti-PHF8 (Bethyl #A301-772A, 1:1000), Anti-phospho-RPA32 (S4/S8) (Bethyl #A300-245A-M, 1:1000), Anti-phospho-RPA32 (Thr21) (GeneTex #GTX130432, 1:1000), Anti-RPA32 (Thermo Fisher #PA5-22256, 1:1000), Anti- RNA polymerase II Ser2P (Abcam #ab5095, WB,1:1000), Anti- RNA polymerase II Ser5P ((3E8) Active motif #61085, 1:1000), Anti-RNA polymerase II (CTD4H8) (Santa Cruz #sc-47701, 1:1000), anti-mouse HRP (Millipore #12-349, 1:10000). Protein expression was visualized using an enhanced chemiluminescence detection system (GE Healthcare, Little Chalfont, United Kingdom) and quantified using a ChemiDoc gel documentation system (Bio-Rad).

## RNA extraction and RT-qPCR

Total RNA from cultured MuSCs was isolated using Trizol reagent (Invitrogen) according to the manufacturer's protocol. 1 μg purified RNA was subjected to reverse transcriptase reaction in the presence of 25 ng/ml random primers and 2.5 mM dA/C/G/TTP with 10 U/ml SuperScript II Reverse Transcriptase (Invitrogen). Primers used for RT–qPCR in Supplementary Data 1.

## S9.6 Dot blot

Genomic DNA was extracted by phenol-chloroform extraction using phase lock tubes according to the manufacturer's protocol. Precipitated DNA was gently spooled and washed with 70% ethanol without centrifugation. DNA was air-dried and re-suspended on ice in 130 μl TE buffer. DNA from each samples was spotted on nitrocellulose membrane using Bio-Dot apparatus (Bio-Rad) and vacuum suction. Membranes were then UV-crosslinked (0.12 J/m2), blocked in 5% milk/TBST, and incubated overnight at 4 °C with mouse S9.6 (Kerafast #ENH001,1:500). Blots were washed 3 times with TBST and incubated with secondary antibody (1:10,000 goat anti-mouse HRP) at RT for 1 h. For total DNA control, DNA with and without RNase H treatment were denatured in 1 M NaOH, 25 mM EDTA at 95 °C for 10 min followed by neutralization with ice-cold 2 M ammonium acetate (pH 7.0). Denatured DNA was spotted onto the nitrocellulose membrane and stained with 0.02% methylene blue in 0.3 M sodium acetate (pH 5.2).

## CUT&RUN and data analysis

CUT&RUN was performed as described in the published protocol[62]. In general, in vitro cultured muscle stem cells (MuSCs) were immobilized on Concanavalin A-coated magnetic beads (Bangs Laboratories), permeated with 0.05% Digitonin (EMD Millipore) and incubated with H4K20me1 (Abcam #ab9051), Pol II S2P antibodies (Abcam #ab5095) or rabbit IgG (Diagenode #C15410206) with 1:100 dilution at 4 °C overnight on a rotator. After wash, the beads were incubated with home-made pA-MNase (0.3 ng/μl final concentration) at room temperature for 1 h and DNA was cleaved by incubation with buffer containing 10 mM $CaCl_2$ and 3.5 mM HEPES pH 7.5 at 0 °C for 20 min. Under this high-calcium/low-salt condition, target cleaved DNA fragments were released and then extracted using phenol-chloroform. DNA fragments were further purified by binding to 1.8 × volume of magnetic carboxylated beads followed by two washes with 85% ethanol. Illumina sequencing libraries were generated using Low Input Library Prep Kit v2 (Takara) following the manufacturer's instructions. Sequencing was performed on the NextSeq500 instrument (Illumina) using v2 chemistry, resulting in minimum of 28 M reads per library with 1 × 75 bp pair end setup.

Raw reads were assessed for quality, adapter content and duplication rates with FastQC 0.11.8 (Andrews S. 2010, FastQC: a quality control tool for high throughput sequence data. Available online at: http://www.bioinformatics.babraham.ac.uk/projects/fastqc). Trimmomatic version 0.39 was employed to trim reads after a quality drop below a mean of Q15 in a window of 5 nucleotides[63]. Only reads longer than 15 nucleotides were cleared for further analyses. Trimmed and filtered reads were aligned versus the Ensembl mouse assembly mm10 (GRCm38 release 99) using STAR 2.7.3a retaining only unique alignments[64]. Reads were deduplicated using Picard 2.21.7 (http://broadinstitute.github.io/picard/) when working with next generation sequencing data in BAM format to mitigate PCR artefacts leading to multiple copies of the same original fragment. Count matrices was produced similarly based on the Spike-in normalized coverage of all mouse genes per sample. All genes located on chromosomes X or Y were excluded from the analysis. Differential genes were identified with a normalized count >50, log2 fold change < −0.5 or >0.5, and multiple testing adjusted P-Value (padj) <0.1 as determined by DEseq2[65].

Venn diagram was generated using BioVenn 1.1.1 (R package)[66]. Identified DEGs were uploaded to the online software DAVID for KEGG pathway analyses[67,68]. Coverage heatmaps and profile plots are based on Spike-in normalized BigWig files produced using Deeptools were generated by deepTool2[69]. Genomic tracks of sequencing data were visualized using pyGenomeTracks[70].

## DRIP-Seq and data analysis

DRIP-seq was optimized based on published protocol[34,71]. In brief, $3 \times 10^6$ MuSCs were resuspended in 1.6 ml of TE buffter with additional 42 μl of 20% SDS and 5 μl Proteinase K (20 mg/ml, Roche) and incubated at 37 °C for 12 h. Genomic DNA prepared as described in dot blot was sheared in 6 × 16 mm microtubes (Covaris, Woburn, MA) to a peak fragment size of 300 bp by sonication on a Covaris machine (E220 evolution) using SonoLab 7.3 software. For each immunoprecipitation, 4.4 μg of sonicated DNA was used. For RNase H treatment, 8 μg of sonicated DNA was digested with 5 μl RNase H (New England Biolabs) in 1x RNase H digestion buffer at 37 °C overnight and purified using standard methods described above. 4 μg of DNA was incubated with 10 μg of S9.6 antibody (Kerafast #ENH001) in binding buffer (10 mM NaPO4 pH 7, 140 mM NaCl, 0.05% Triton X-100) overnight at 4 °C, and DNA-protein complex was incubated with Protein A Dynabeads (Thermo Fisher Scientific) for at room temperature for 2 h. After 3 times washes with binding buffer, DNA was eluted after incubation with elution buffer (50 mM Tris pH 8, 10 mM EDTA, 0.5% SDS, Proteinase K) for 45 min at 55 °C. DNA was purified as described above and DNA libraries were synthesized from ssDNA using the Accel-NGS 1 S DNA library kit (Swift Biosciences, Ann Arbor, MI) according to the manufacturer's protocol. Using multiplexing adapters from the 1 S Plus Set A Indexing Kit (Swift Biosciences), adapter-ligated DNA was amplified by PCR, then size selected using a left/right AMPure XP size selection (Beckman Coulter). Library DNA was analyzed on a Bioanalyzer DNA HS (Agilent, Santa Clara, CA), quantified by qPCR using NEBNext Library Quant Kit for Illumina (New England Biolabs), and sequenced on a HiSeq 4000 (Illumina, San Diego, CA) at the Stanford Genome Sequencing Service Center, using 2 × 151 bp sequencing.

Raw data were processed similarly to the CUT&RUN data. Peaks were called comparing the respective treatment to Input samples using Epic2 0.0.41[72]. Peaks overlapping ENCODE blacklisted regions (known misassemblies, satellite repeats) were excluded. To enable comparison of peaks in different samples, the resulting lists of significant peaks were overlapped and unified. Sample counts for union peaks were produced using bigWigAverageOverBed (UCSC Toolkit). Reads mapping against Spike-in organism Drosophila melanogaster (BDGP6) were counted for each sample and used to normalize the matrix of union peak scores. Union peaks were annotated with the gene having the longest overlap based on reference data of Ensembl release 99. All peaks located on chromosomes X or Y were excluded from the analysis. Differential peaks/genes were identified with a normalized count >30, fold change < −1.5 or >1.5, and multiple testing adjusted P-Value (padj) <0.1 as determined by DEseq2. Peaks were annotated with UROPA[73] based on the overlap of the peak center using the following priority of feature types: (1) promoter (TSS ± 1 kb), (2) TES region (TES + 3 kb downstream), (3) gene, (4) intergenic. If multiple genes of the same feature type were intersected, the one with the longest overlap was selected. The genomic background represents the distribution of features if reads were randomly placed.

## PRO-seq library preparation and data analysis

Nuclear run-on assays with nuclei isolated from proliferating *Ctrl* and *Kmt5b*<sup>sKO</sup> MuSCs were performed and sequencing libraries were constructed as described previously[74]. In brief, $1 \times 10^7$ nuclei were incubated with the same volume of 2-Biotin Nuclear Run-On (NRO) reaction mixture (10 mM Tris-HCl pH 8.0, 300 mM KCl, 1% Sarkosyl, 5 mM MgCl2, 1 mM DTT, 500 μM biotin-11-C/UTP (Perkin-Elmer), 5 mM A/GTP, 0.8 u/μl RNase inhibitor) for 3 min at 30 °C. Nascent RNA was extracted using Trizol and fragmented by base hydrolysis in final 25 μl 0.2 N NaOH on ice for 10-12 min followed by neutralization with 1x volume of 1 M Tris-HCl pH 6.8. Fragmented nascent RNA was cleaned up using P-30 column (Bio-rad) and immobilized onto the Streptavidin M-280 magnetic beads (Invitrogen) following the manufacturer's instructions. The beads were washed with high salt (2 M NaCl, 50 mM

Tris-HCl pH 7.4, 0.5% Triton X-100), medium salt (300 mM NaCl, 10 mM Tris-HCl pH 7.4, 0.1% Triton X-100), and low salt (5 mM Tris-HCl pH 7.4, 0.1% Triton X-100), successively. Bound RNA was extracted from the beads using Trizol (Invitrogen) in two consecutive extractions, and the RNA fractions were pooled, followed by ethanol precipitation. For the first ligation reaction, fragmented nascent RNA was dissolved in H2O and incubated with 10 pmol of reverse 3′ RNA adaptor (5′p-rGrArUrCrGrUrCrGrGrArCrUrGrUrArGrArArC rUrCrUrGrArArC-/3′InvdT/) and T4 RNA ligase I (NEB) under manufacturer's condition for 6 h at 20 °C. Ligated RNA with biotin-labeled 3′ ends was purified by binding to Streptavidin bead. After extraction with Trizol, 5′ ends of the RNA were repaired with Tobacco Acid Pyrophosphatase (TAP, Epicentre) and Polynucleotide Kinase (PNK, NEB), then was ligated to reverse 5′ RNA adaptor (5′-rCrCrUrUrGrGrCrArCrCrCrGrArGrArArUr-UrCrCrA-3′). Ligated RNA products were further purified through binding to streptavidin beads followed by extraction with Trizol. Adaptor ligated nascent RNA was reverse transcribed using 25 pmol RT primer 5′-AATGATACGGCGACCACCGAGATCTACACGTTCAGAGTTCT ACAGTCCGA-3′ for TRU-seq barcodes (RP1 primer, Illumina).

A portion of the RT product was used for trial amplifications to determine the optimal number of PCR cycles. Final amplification was performed by NEBNext High-Fidelity 2x PCR Master Mix (NEB). Based on amplification curve 7–9 additional cycles were used after 5 initial cycles. Final libraries were cleaned up by 1x SPRI bead purification, followed by additional 2-sided bead cleanup to eliminate unwanted small and big fragments. Sequencing was performed on the NextSeq500 instrument (Illumina) using v2 chemistry, with minimum of 28 M reads per library and 1 x 75 bp pair end setup.

Raw reads were assessed for quality, adapter content and duplication rates with FastQC 0.11.8 (Andrews S. 2010, FastQC: a quality control tool for high throughput sequence data. Available online at: http://www.bioinformatics.babraham.ac.uk/projects/fastqc). Trimmomatic version 0.39 was employed to trim reads after a quality drop below a mean of Q15 in a window of 5 nucleotides[63]. Only reads longer than 15 nucleotides were cleared for further analyses. Trimmed and filtered reads were aligned versus the Ensembl mouse assembly mm10 (GRCm38 release vM15) using STAR 2.6.1d retaining one random alignment in case of multimapping reads[64]. Reads were deduplicated using Picard 2.18.16 (http://broadinstitute.github.io/picard/) when working with next generation sequencing data in the BAM format to mitigate PCR artefacts leading to multiple copies of the same original fragment. Target features were produced by extracting all isoforms of protein-coding genes of Ensembl GRCm38 release vM15 with length > = 1000 nt and creating for each: (1) TSS region (start = TSS, stop = 400 nt downstream of TSS), (2) body region (start = 400 nt downstream of TSS, stop = TES). Read counts for TSS/body features were created using bigWigAverageOverBed (UCSC Toolkit) by only counting reads from the correct strand. Count matrices for TSS/body features were TPM normalized (sequencing depth, feature length) and combined. All isoforms with an average TPM < 0.5 in either TSS or body features were removed. One isoform was selected for each gene as the primary transcript based on the largest average TPM score on the TSS feature. The pausing index ratio (PI) was computed as TPM TSS/TPM Genebody. Sequencing depth normalized BigWig files were produced using Bedtools 2.29[75] and wigToBigWig (UCSC Toolkit). eRNAs were predicted using the NRSA v2 pipeline which locates nascent bidirectional transcripts[76]. Volcano plot was created in R with ggplot2 package. All eRNAs/genes located on chromosomes X or Y were excluded from the analysis. eRNAs (fold change > 0.5 or < −0.5 and FDR < 0.1) was colored with significant change. Coverage heatmaps and profile plots are produced using Deeptools were generated by deepTool2[69]. Genomic tracks of sequencing data were visualized using pyGenomeTracks on Galaxy platform (https://usegalaxy.eu/)[70].

## ChIP seq data analysis

Mouse wild type MuSC samples of H3K4me1 ChIP-Seq project PRJNA412267 were downloaded from the European Nucleotide Archive. Samples were processed similarly to the CUT&RUN data. Peak calling was perfomed with MUSIC[77]. DESeq2 was used to normalize the resulting matrix of union peak scores.

## Human cancer dataset analysis

Copy number segments (After remove germline cnv) and copy number–gene level (gistic2_thresholded) from TCGA Pan-Cancer (PANCAN) cohort were downloaded from UCSC Xenahub (https://tcga.xenahubs.net). DNA methylation (Methylation450K) data, batch effects normalized mRNA data and ssGSEA analysis of 1387 constituent PARADIGM pathways were obtained from TCGA PancanAtlas Xenahubs (https://pancanatlas.xenahubs.net). The primary tumor samples from SARC, THCA, PRAD, LUAD and BRCA were selected for analysis and visualized by Xenabrowser and Prism8. Clinical data of cancer patients related to Supplementary Fig. 8 can be found in Supplementary Data 2. Information about probes for KMT5B regions in the Illumina Infinium HumanMethylation450 (450K) BeadChip array is provided in Supplementary Data 3.

## Statistics and reproducibility

Animal studies were performed without blinding and no animals were excluded from the analysis. All experiments were repeated at least three times with similar results. Sample size for in vitro studies was chosen based on observed effect sizes and standard errors. Significant testing were performed using the GraphPad Prism 8.0 (GraphPad Software) program to determine statistical significance between groups. Data are represented as mean ± the standard error of the mean. Two-tailed *P* values for the unpaired Student's *t*-test and adjusted *p* values for one-way ANOVA and two-way ANOVA are indicated in the graphs. One-way and two-way ANOVA analysis was done with Tukey's test for correction of multiple testing. *P* values lower than 0.05 were considered statistically significant. No statistical method was used to predetermine sample size.

## Reporting summary

Further information on research design is available in the Nature Portfolio Reporting Summary linked to this article.

## Data availability

The data that support this study are available from the corresponding auhtors upon request. The CUT&RUN, PRO-seq and DRIP-seq data are available via the GEO accession numbers GSE173259, GSE173257 and GSE173258, respectively. Source data are provided with this paper.

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

## Acknowledgements

This work was supported by the Max Planck Society, the German Research Foundation Transregional Collaborative Research Centre SFB-TR81 (TP02), the Transregional Collaborative Research Centre SFB267 TP A05, the Collaborative Research Centre SFB1213 TP B02, Excellence Cluster Cardio-Pulmonary Institute (CPI), the LOEWE project iCANx, and the German Center for Cardiovascular Research. We would like to thank Ann Artzberger and Kikhi Khrienovo for support with FACS sorting. We also would like to thank Dr. Stephan Hamperl for sharing detailed protocols for DRIP-seq and Dr. Ronald Wong for sharing protocols for the DNA fiber spreading assay.

## Author contributions

T.Z., S.G., D.D., X.G. performed experiments, T.Z. and K.C. performed bioinformatics analysis of sequencing data. T.Z., X.Y. and T.B. interpreted data, T.Z., X.Y. and T.B. wrote and edited the manuscripts, T.Z., Y.Z., X.Y. and T.B. conceived the project.

## Funding

## Competing interests

The authors declare no competing interests.
