## [Peer Review File · Nature Communications]

Replication collisions induced by de-repressed S-phase transcription are connected with malignant transformation of adult stem cellsREVIEWER COMMENTS

Reviewer #1 (Remarks to the Author):

The authors show that loss of Suv4-20h1 in muscle stem cells (MuSCs) results in increased H4K20me1 and increased transcription of many genes during S phase. Increased transcriptional activity appears to result in enhanced transcription replication collisions (TRCs) and enhanced DNA damage in Suv4-20h1 ko MuSCs. Augmented ATR signaling connects to p53 activation and senescence induction in Suv4-20h1 deficient MuSCs. Interestingly, loss of p53 in these cells promotes rhabdomyosarcoma formation.

Overall, the findings are very interesting and put Suv4-20h1 and H4K20me1 as important players in cell cycle regulated gene expression. In addition, the discovery of RMS formation in Suv4-20h1/p53 mice provides an interesting basis to investigate the role of SUV4-20H1 in human tumorigenesis.

The following points need to be addressed before publication:

- 1) Figures 2c,d and Expanded view 2e,f: It is not clear what the p values indicate here.
- 2) Figure 2k shows pathway analysis of 8020 Suv4-20h1 target genes (K20me1 up / PolII up) For so many genes a pathway analysis is quite meaningless and should be removed. It is not needed to make important arguments.
- 3) The analysis of transcriptional changes in Suv4-20h1 ko cells is focused on enhancer RNAs (Figure 3). It would be important to show a similar analysis for gene transcription.
- 4) Figure 4. The DRIP-seq analysis appears to be connected with very high background and consequently uncertain peak detection. Overall it seems that R loops increase based on Figure 4B, but the subsequent analysis is uncertain. Ultimately, R loop analysis is not so critical for the overall argumentation, so it would be easy to remove it. Alternatively, a more robust analysis of peak detection with replicate experiments could be shown.
- 5) line 331: We found that loss of Suv4-20h1 leads to accumulation of H4K20me1 specifically in S-phase.

The increase in H4K20me1 is not specific to S phase, as shown in figure 2B. Cells outside S phase also feature high H4K20me1 levels. This makes full sense if assuming that Set8 induces H4K20me1 on all new nucleosomes and Suv4-20h1 being important for me2 conversion in genes. The statement needs to be amended.

- 6) TCGA data analysis in Figure 7 is suggestive of a role of SUV4-20H1 in human tumors. However, in absence of any validation experiments (at least SUV4-20H1 and/or H4K20me1 analysis in tumor samples) the figure is a bit speculative. I would recommend moving the figure to the supplement and to tone down the conclusions and the last paragraph in the discussion.

Reviewer #2 (Remarks to the Author):

In the manuscript entitled "De-repressed S-phase transcription induces replication collisions causing genomic instability and malignant transformation of adult stem cells", Zhang et al. describe a role for the methyltransferase SUV4-20H1 in preventing transcription-replication conflicts (TRC) in murine muscle stem cells. They show that cells lacking Suv4-20h1 (SH1SKO cells) exhibit reduced global H4K20me2 and an accumulation of H4K20me1 histone modifications. They correlate these modifications with increased transcription during S-phase and enhanced formation of TRC-induced R-loops and DNA damage in SH1SKO cells. They show that loss of p53 in these cells overrides DNA-

damage induced senescence and leads to malignant transformation and the development of rhabdomyosarcoma in mice. The authors also uncover that human sarcomas often harbour a low expression of SUV4-20H1 which correlates with poor patient outcome. The authors put forth a model in which SUV4-20H1 acts as a key tumour suppressor in sarcomas by repressing S-phase transcription, TRCs and associated DNA damage.

The authors employ a variety of approaches to support their hypothesis, employing genome-wide analyses combined with cell biology assays and in vivo studies. The manuscript is well-structured and easily understood, although sometimes the figures are too data-dense. Below are some general points for considerations to the authors:

1. The authors have well-designed experiments to answer their specific questions and put forth multiple interesting observations, however, some mechanistic links are missing and are worth re-considering. For example: the authors interpret observations in Suv4-20h1 KO as resulting from H4K20m1 increased levels. Although lack of Suv4-20h1 indeed results in higher H4K20m1 levels as demonstrated by the authors, other effects of this KO, not attributable to this histone mark, could not be excluded. It would be beneficial if the authors modulate the levels of other histone modifiers, such as SETD8 to control for some of their readouts, which would also serve as a control for the specificity of the role of Suv4-20h1. Alternatively, some complementation experiments with a catalytically-dead Suv4-20h1 would provide the specificity of this enzymatic function to their observed phenotypes. In light of the authors using one cell model to support their findings, such control experiments are important.

2. The authors make a prominent point of how their findings in mouse cells apply to human sarcomas by extracting genomic data from available datasets and showing matching correlations (Fig. 7). While this is interesting, extrapolating their hypothesis to human cells requires at least recapitulating some of their main findings in human sarcoma cells. This is an important aspect for the suitability of the manuscript for a wider audience.

3. It is important that the authors show cell cycle profiles of SH1SKO cells and if there are significant differences. A skew in cell cycle phase, for example, could account for observations the author report for whole-cell populations such as their western blots of DNA damage markers and histone modifications (Fig.1g, Fig.2a, Fig3d). In such cases, using synchronized cell populations and cell cycle-specific analyses will provide a more accurate representation of the effects observed.

More specific comments with regards to the experiments:

4. Figure 1b: The authors use increases 53BP1 levels as a measure of DNA damage, although it is known that 53BP1 is recruited to DNA damage via H4K20me2 and a reduction in that mark would in fact decrease 53BP1 recruitment. The authors should address this discrepancy more carefully (line 335 they mention that there is no change in DNA damage response).

5. Figure 2: The authors make a point about changing levels of H4K20me1 based on immunofluorescence images (Fig. 2c), but show no quantification or statistical analysis to back up these claims (also in supplementary data Fig. 2a,b,c). Similarly, in Figure 2c, it is not clear what threshold the authors used to quantify the cells as pATR-positive, as the images show definitely more than the 10% in the accompanied quantification. It is not clear why the analysis of p-RPA is not done in only EdU+ve cells, similar to other IF analyses as pRPA is also present at DNA double-strand breaks (and in G2 cells) and not only stalled forks. There is no quantification for this panel (Fig. 4d). In the case of DNA damage, γ H2AX foci quantification in multiple figures (Fig. 1g, Fig. 5a) in EdU+ve cells is a more sensitive, accurate measurement than total WB signal.

6. Figure 4: The PLA assay is really interesting and offers convincing data regarding the presence of TRC in SH1SKO cells (Fig 4a). However, there are concerns regarding the specificity of the foci observed, especially that they are present in EdU-ve cells as well and if the authors have controlled for this (for example by depleting PCNA) or how do the authors explain the presence of PLA signal in these cells? Additionally, the authors also make a statement about PLA signal being found mainly in euchromatin (line 188), with no supporting data other than one IF image. Did the authors have quantify this?

7. Figure 7: the authors make a point of increased ATR activation in sarcomas and compare it to other tumors but do not explain their choice of these tumors. Perhaps it is beneficial to include a group of tumors that are known to exhibit replication stress- associated ATR activation such as ovarian

carcinomas, as positive control to their GSEA score. This is also predicted to be the case to at least for some BRCA-deficient breast cancer tumors, but it is not clear how the tumor data sets were selected and why no such increase is observed.

Reviewer #3 (Remarks to the Author):

Previously, this group showed that KMT5B/SUV420H1 was necessary to maintain muscle satellite cell quiescence. They extend this finding in the current study to show that KMT5B/SUV420H1 is necessary to prevent S-phase transcription, transcription-replication-collisions (TRC), and tumor formation. In this study they show that deletion of KMT5B/SUV420H1 (which the authors abbreviate as SH1) in MuSCs results in signs of DNA double strand breaks (53BP1 foci, phospho-gH2AX foci, aberrant mitoses, increased p53 and p21) together with decreased cell expansion. This correlated with decreased H4K20me2 and increased H4K20me1 overall and specifically increased me1 in S-phase cells, which together with a correlation of increased me1 and PolIIIS2P in gene bodies and enhancer elements and increased EU incorporation in PCNA+ cells indicated that the enhanced transcription of these genes was present in S-phase. The authors use several approaches to show that this results in TRCs: proximity ligation assays (PLA) between PCNA and PolII together with EDU labeling, S9.6 Ab on dot-blots or together with MyoD antibody for IF or in ChIP, detection of phosphor ATR and RPA32, and reduction of gH2AX. PLA, and S9.6 foci with low dose DRB treatment. The analysis of the S9.6 ChIP showed enrichment of genes amplified or re-arranged in rhabdomyosarcoma and other cancers. Deletion of p53 rescued the proliferation defect of SH1 deficient cells and resulted in tumors during muscle regeneration in double-KOs (DKs), which had histologic characteristics of rhabdomyosarcomas. TCGA and other analyses showed prevalence of 11q and SH1 loss in sarcomas, as well as decreased SH1 and p53 RNA associated with elevated ATR and decreased ATM pathways.

Overall, this is a well done and well presented study.

The study convincingly shows increased S-phase transcription and TRCs with the acute total loss of SUV420H1/KMT5B, and tumor formation when coupled with acute loss of p53. The extension to human cancers has many important correlative findings that support a similar role in human sarcomas and other cancers. A major concern is that the current presentation of the data embraces the authors model and leaves the reader to identify additional considerations that might temper such an enthusiastic embrace. Although listed as major concerns below, these can mostly be addressed by changes in the text or minor experimental extensions.

Major concerns:

1. The mouse data nicely shows the TRCs and DNA damage associated with loss of KMT5B, as well as tumor formation when combined with p53 loss in a model of muscle regeneration. Genome instability, specifically re-arrangements and CNVs, is not directly shown in the resulting tumors and would strengthen the model. Showing this might be necessary to maintain the wording for the major conclusions. The title, abstract, and text makes the major conclusion that the de-repressed S-phase transcription causes genome instability. Is the demonstration of TRCs and gH2AX sufficient to conclude genome instability without a direct demonstration of re-arrangements or CNVs? Not for me, but I understand there might be some disagreement. Still, the understanding of the general reader should be considered and either wording for major conclusions changed and/or transparent definitions in the context of stating the major conclusions.
2. The tumor formation in genetic models and in human data remains correlative. The authors nicely showed that loss of KMT5B resulted in TRCs, and that loss of KMT5B (or, in the case of human cancers, diminished KMT5B) was associated with tumors, but the study did not show that preventing TRCs in the face of KMT5B loss prevented the cancers and leaves open the possibility that other consequences of KMT5B loss might be the cause of transformation. The title and text is less nuanced

and claims that de-repressed S-phase transcription causes malignant transformation. This concern can be addressed with either additional experimental studies, such as a rescue of tumor formation by preventing TRCs, or more careful wording.

3. The generation of rhabdomyosarcoma, as opposed to other tumor types, in a mouse model that disrupts KMT5B in MuSCs seems directly related to the model system. To suggest specificity for RMS in the mouse model would require comparisons with deletions in other tissues at similar stages of tissue regeneration. Again, well beyond the scope of this study, but well within what needs to be more critically presented, particularly since this is used as a segue to human cancers and a focus on sarcomas.

4. The mouse models relied on an acute complete deletion of KMT5B and did not assess heterozygotes, whereas the human cancers appear to have decreased expression but not necessarily complete loss. Does haploinsufficiency lead to TRCs? Maybe not necessary to experimentally address, but necessary to discuss.

5. Discussion of therapeutic implications in the discussion would be of interest. This might also address whether the TRC induced by KMT5B loss is transient or persistent in the mouse models, e.g., do the mouse tumors show continued TRC? It would also be interesting to address whether RMS cell lines show TRC that can be rescued by DRB. Again, maybe beyond the scope of this study, but a point that should be either addressed experimentally or in the Discussion.

6. Figure 7A shows a heat plot for the methylation beta value where a higher number should reflect a higher percentage of methylation. The figure shows lower beta values in the promoter region of sarcomas, the text indicates that these lower values mean higher methylation; whereas panels e and f show higher values for the sarcomas (not specified whether these are beta values) indicating higher methylation. Maybe I am misunderstanding, but something seems inconsistent.

Minor points:

7. Should the authors use the official gene and protein name, KMT5B?

8. Page 8 introduces the term TCR as compared to TRC, likely an error?

9. Page 11 line 285 classifies the mouse tumors as embryonal rhabdomyosarcoma. This seems to go beyond the data or even the ability to classify genetic mouse tumors with a human classification system.

Reviewer #4 (Remarks to the Author):

The manuscript 'De-repressed S-phase transcription induces replication collisions causing genomic instability and malignant transformation of adult stem cells' by Zhang et al., investigates the effects of Suv4-20h1 loss on transcription-replication conflicts in adult muscle stem cells. This is generally an excellent, well-thought-out and executed study with wide-ranging implication for the general understanding of how transcription-replication conflicts occur and the potential involvement of Suv4-20h1 in the formation of rhabdomyosarcomas.

Despite my sentiments I do have some minor and major comments that in my opinion are essential before publication.

Major

1. On page 5 lines 100-101 the authors write: 'We concluded that loss of Suv4-20h1 causes DNA damage and replication stress in proliferating SH1sKO MuSCs both in vitro and in vivo.' Increased

53BP1, γ H2AX, micronuclei and aneuploidy imply replication stress however replication stress is defined by DNA synthesis slowing and/or replication fork stalling (PMID: 25907220). Therefore, the author must show DNA synthesis slowing and/or replication fork stalling using the DNA Fibre assay in MuSC cells. Furthermore, this statement would be better suited at the end of the next paragraph (circa lines 112/113)

2. On page 6 lines 128-130 the authors discuss Genome-wide deposition of H4K20me1 shown in figure 2B. These images only show several individual cells without any quantification and thus do not provide any evidence for the H4K20me1 accumulates in S-phase. The author must provide quantification of EdU (+/-) and H4K20me1 (+/-) similar to data shown in figure 3E. In line with these changes the authors must modify the discussion accordingly if this data is not provided (page 12 lines 331-332).

3. The data shown in Figure 5 adequately demonstrates repression of transcriptional elongation reduces R-loop formation (and potentially genomic instability). However, the authors have provide little evidence for the effects on transcription-replication conflicts. In figure 5A, the γ H2AX quantification and western blots do not correlated and are not convincing. The authors need to repeat experiments shown in Fig 1B-F looking at either 53BP1 or γ H2AX in EdU+ cells +/- DRB.

4. I have major reservation about the interpretation of the R-loop data in this manuscript. On page 8 lines 190-191 the authors write: 'In case of head-on (HO) orientations, TRCs replication forks will stall and promote formation of genome-destabilizing R-loop.' Firstly, the grammar in this sentence needs addressing. Secondly the authors here and in accompanying statements throughout the manuscript, are suggesting head-on (HO) transcriptional-replication conflicts cause R-loop formation. This may well be true however R-loops can be generated in numerous ways either physiologically (e.g. transcription or DNA damage repair) and pathologically (excessive transcription following oncogene activation). The experiments discussed in fig 4 or Extended Data Fig. 4 do not demonstrate head-on-collisions between replication or transcription machinery. The data only shows increased R-loop formation following the loss of SUV420H1. On page 8 lines 204-205 the authors also write: 'To determine precisely the locations of TCR-induced R-loops....' Again the authors have not provided any evidence of the 'type' of transcription-replication conflicts or R-loops were generated following TRC. Therefore, the authors are over stating their findings (also in the discussion) based on a false premise.

The authors need to delete any reference to head-on-collisions or that R-loops are generated following collision of the replisome and transcriptional machinery. For example the following statement must be amended (page 10 lines 249-252): 'Taken together, these data indicate that increased transcription activity of a large group of coding genes and eRNAs during S-phase promotes collision of the replisome and transcriptional machinery leading to R-loop formation and genome instability of SH1sKO MuSC (Fig.5g).'

On page 8/9 line 209-212, the authors contradict their previous premise: '64.2% (1508 from 2350) of R-loop containing genes showed increased H4K20me1 and Pol II S2P enrichment, indicating that enhanced transcriptional activation is the main cause for increased R-loop formation in SH1sKO MuSC.' Are R-loops caused by head-on-collisions or enhanced transcriptional activation? In my opinion the latter, since 64.2% (1508 from 2350) of R-loop containing genes showed increased H4K20me1 and Pol II S2P enrichment and DRB treatment reduces R-loop formation! Finally, in order to show Increased R-loop formation causes transcription-replication conflicts the authors would be required to assess DNA damage (RPA, 53BP1 or γ H2AX in EdU+ cells at a minimum) in cells with or without RNaseH1 (see PMID: 27725641).

Minor

1. in the introduction on page 3 line 65, the abbreviation MuSC needs defining.

2. On page 5 line 97 the authors write: 'Importantly, a marked increase of 53BP1 nuclear bodies (NB) was apparent in the G1 phase of SH1sKO MuSC, suggesting failure to complete DNA replication during the last cell cycle (Fig.1c).' It is not evident how the G1-phase or 2N DNA content was identified. Authors must provide this information.

3. In the methods on page 17 line 481 the title '9.6 Dot blot; needs changing to S9.6 Dot blot.
4. On page 9 line 224, Figure references needs amending.
5. On page 9 line 236, the author must provide details of how long cells were treated with DRB.
6. On page 10 CTX abbreviation needs defining.
7. On page 14 the sentence on lines 371-372 needs some context.
8. The y-axis in Fig 6G needs amending.

REVIEWER COMMENTS

Reviewer #1:

The authors show that loss of Suv4-20h1 in muscle stem cells (MuSCs) results in increased H4K20me1 and increased transcription of many genes during S phase. Increased transcriptional activity appears to result in enhanced transcription replication collisions (TRCs) and enhanced DNA damage in Suv4-20h1 ko MuSCs. Augmented ATR signaling connects to p53 activation and senescence induction in Suv4-20h1 deficient MuSCs. Interestingly, loss of p53 in these cells promotes rhabdomyosarcoma formation.

Overall, the findings are very interesting and put Suv4-20h1 and H4K20me1 as important players in cell cycle regulated gene expression. In addition, the discovery of RMS formation in Suv4-20h1/p53 mice provides an interesting basis to investigate the role of SUV4-20H1 in human tumorigenesis.

Response: We thank the reviewer for the positive evaluation of the manuscript and for the helpful comments.

The following points need to be addressed before publication:

1) Figures 2c,d and Expanded view 2e,f: It is not clear what the p values indicate here.

Response: We apologize for omitting a detailed description of p-values. The p-values in Figures 2c, d and Expanded view 2e, f describe the significance of the Spearman's Rank Correlation test. The r-values indicate the strength of a link or relationship between two sets of data with $r=1$ indicating perfect ranking association, whereas the p-values indicate the significance of the correlation between two conditions. In our case the p-values (<0.0001) reveal that both the 'positive correlation' of H4K20me1 and Pol IIS2P occupancy in gene bodies of coding genes (Fig. 2c, d) and the 'no correlation' in gene bodies of rRNAs, snRNAs, snoRNAs, and scaRNAs (Original Supplementary Fig. 2e, f; revised Supplementary Fig. 2j, k) are not caused by random sampling but are highly significant. We have now defined the p-values in the figure legends of revised Fig. 2 and Supplementary Fig. 2.

2) Figure 2k shows pathway analysis of 8020 Suv4-20h1 target genes (K20me1 up / Pol II up) For so many genes a pathway analysis is quite meaningless and should be removed. It is not needed to make important argments.

Response: The reviewer is right. We have removed the pathway analysis as recommended.

3) The analysis of transcriptional changes in Suv4-20h1 ko cells is focused on enhancer RNAs (Figure 3). It would be important to show a similar analysis for gene transcription.

Response: We followed the reviewer's advice and analyzed enrichment of the PRO-seq signals at the TSS and within the gene bodies of the 8020 genes that showed both upregulated H4K20me1 content and increased Pol IIS2P occupancy after *Kmt5b* depletion in *Ctrl* and *Kmt5b^{SKO}* MuSCs. We detected increased PRO-seq signals both at TSS and over gene bodies in *Kmt5b^{SKO}* MuSCs, suggesting that the transcriptional activities of these genes were upregulated, which is consistent with the pausing index results (revised Supplementary Fig. 3a, b).

4) Figure 4. The DRIP-seq analysis appears to be connected with very high background and consequently uncertain peak detection. Overall it seems that R loops increase based on Figure 4B, but the subsequent analysis is uncertain. Ultimately, R loop analysis is not so critical for the overall argumentation, so it would be easy to remove it. Alternatively, a more robust analysis of peak detection with replicate experiments could be shown.

Response: We agree that the genome browser tracks of DRIP-seq signals across the *Rb1* gene locus showed a relative high background (old Fig. 4k). The genome browser tracks of DRIP-seq signals indeed showed relatively high background across the *Rb1* gene locus, which we assume is an odd feature of the locus and not representative for the dataset (old Fig. 4k). We re-evaluated our DRIP-seq data carefully based on three biological replicates using the same peak calling method as before. The Spearman correlation matrix (revised Supplementary Fig. 4g) indicates that the normalized union peak counts are highly reproducible among the biological replicates, arguing against random or background peak detection. We now show DRIP-seq peaks within three oncogenic genes including *Rasa1*, *Agap2* and *Arid4b* in *Kmt5b*^{SKO} MuSCs. The selection is representative for the whole data set, replacing the genome browser tracks of DRIP-seq signals in the *Rb1* gene locus (revised Fig. 4k; Fig.1 for the reviewer).

Figure 1 for the reviewer: Genome tracks for DRIP-seq of *Rasa1*, *Agap2* and *Arid4b* genes in three replicates of *Ctrl* and *Kmt5b*^{SKO} MuSCs.

5) line 331: We found that loss of *Suv4-20h1* leads to accumulation of *H4K20me1* specifically in *S*-phase.

The increase in *H4K20me1* is not specific to *S* phase, as shown in figure 2B. Cells outside *S* phase also feature high *H4K20me1* levels. This makes full sense if assuming that *Set8* induces *H4K20me1* on all new nucleosomes and *Suv4-20h1* being important for *me2* conversion in genes. The statement needs to be amended.

Response: We are still convinced that accumulation of *H4K20me1* after inactivation of *Suv4-20h1/Kmt5b* predominantly happens in the *S* phase. We think that we presented misleading representative images and also omitted a proper quantification. Similar comments were also raised by reviewer 2 (comment #5) and 4 (comment #2). Previously, we had used different numbers of z-stack layers from confocal images, which led to the wrong impression. We now quantified the total *H4K20me1* content in *EdU*⁺ and *EdU*⁻ cells (mainly *G1*-phase cells, since cells at *G2/M* phase are excluded judged by *DAPI* intensity), based on epifluorescence microscopy (revised Fig. 2b). The median fluorescent intensity (MFI) of *H4K20me1* in *S*-phase (*EdU*⁺) *Ctrl* MuSC is significantly lower than that in *EdU*⁻ *Ctrl* MuSC. This result fits well with previous findings, demonstrating that *H4K20me1* deposition is lowest during *S*-phase¹. We also show that *Suv4-20h1/Kmt5b* depletion

dramatically increased the signal (MFI) for H4K20me1 in EdU+ MuSC, whereas no increase is apparent in EdU- MuSC, supporting the conclusion that inactivation of *Suv4-20h1/Kdm5b* leads to increased accumulation of H4K20me1 predominantly in S-phase (revised Fig. 2b).

The relative stable H4K20me1 level in *Kmt5b^{SKO}* MuSCs outside the S-phase (cells mainly in G1-phase (EdU-)) compared to the S-phase (EdU+) is probably due to two reasons:

1) As also mentioned by the reviewer, SETD8/KMT5A generates H4K20me1 on new nucleosomes whereas SUV4-20H1/KMT5B is important for conversion of H4K20me1 into H4K20me2. Some conversion of H4K20me1 to me2 occurs indeed during G1-phase, but this conversion is repressed in G2/M phase by unknown mechanism². H4K20me2/3 in G1 phase is inversely correlated with nucleosome turnover in protein-coding genes³, meaning that H4K20me2/3 enriched genes have a low turnover rate of nucleosomes either due to low transcriptional activity or efficient nucleosome recycling. We reason that the low turnover rate of H4K20me2 during G1-phase will prevent strong changes of H4K20me2/3 enriched genes after KMT5B depletion. In contrast, the conversion of H4K20me1 to H4K20me2 needs to be actively and dynamically controlled by KMT5B before DNA replication to avoid transcription replication collisions. Therefore, the changes in H4K20me after *Kmt5b* knockout during G1-phase will be less pronounced compared to changes during S phase. Accordingly, quantification of the signals revealed that changes of H4K20me2 during G1-phase are less dramatic compared to changes in S-phase (revised Supplementary Fig. 2a).

Several publications have demonstrated that newly incorporated nucleosome will remain unmethylated at H4K20 until the end of S phase to inhibit error-prone non-homologous end joining and promote homologous recombination after DNA replication^{4, 5}. Thus, we excluded the possibility that accumulation of H4K20me1 during S-phase of *Kmt5b^{SKO}* MuSC occurs on newly incorporated nucleosome after DNA replication.

2) Reduced activity of SETD8/KMT5A or the presence of a H4K20me1 demethylase such as PHF8 might prevent a higher accumulation of H4K20me1 during the G1-phase in *Kmt5b^{SKO}* MuSCs. Indeed, we observed a modest downregulation of SETD8/KMT5A and an increase of PHF8 at protein level, suggesting the existence of a feedback loop that reduces H4K20me1 on unwanted regions (revised Supplementary Fig. 2d). Since SETD8/KMT5A is degraded when cells enter the S-phase, such compensation can only happen outside the S-phase.

We are convinced that our results support the conclusion that loss of *Kmt5b* leads to accumulation of H4K20me1 predominantly during the S-phase. Nevertheless, we have slightly amended the statement as suggested.

6) TCGA data analysis in Figure 7 is suggestive of a role of SUV4-20H1 in human tumors. However, in absence of any validation experiments (at least SUV4-20H1 and/or H4K20me1 analysis in tumor samples) the figure is a bit speculative. I would recommend moving the figure to the supplement and to tone down the conclusions and the last paragraph in the discussion.

Response: We thank the reviewer for the comment, which motivated us to conduct further studies. We compared the expression of *KMT5B* and H4K20me1 in a rhabdomyosarcoma cell line (RD), in an osteosarcoma cell line (U2OS), and two carcinoma cell lines (MCF7, H1299) with two human myoblast cell lines and a human embryonic kidney cell line (HEK293). Consistent with the TCGA data, expression of *KMT5B* but not *KMT5A* was dramatically decreased in RD cells, while the H4K20me1 level was significantly higher in RD cells than in the other cell lines (revised Supplementary Fig. 9a, b, c). Moreover, we found that copy number loss and promoter

hypermethylation contributes to suppression of *KMT5B* expression in RD cells (revised Supplementary Fig. 9d, e). The low expression of *KMT5B* in RD cells correlated with high numbers of PCNA-Pol II S2P PLA+ foci and R-loop formation, validating the results of the TCGA data analysis that low expression of *KMT5B* is associated with a high rate of transcription replication collisions (revised Supplementary Fig. 9f, g).

We do agree with the reviewer that it is a bit speculative to correlate the expression of *KMT5B* with the ATR signaling pathway in different types of cancer. Loss of *KMT5B* is certainly not the only reason for ATR activation in cancers. Changes in the genome of different cancer cells will have a major impact on the response to transcription replication collisions. Therefore, we have deleted the data on ATR/ATM signaling, moved all the human cancer data to Supplementary Fig. 8, and toned down our statements about the clinical implications of our findings in the discussion.

Reviewer #2:

In the manuscript entitled “De-repressed S-phase transcription induces replication collisions causing genomic instability and malignant transformation of adult stem cells”, Zhang et al. describe a role for the methyltransferase SUV4-20H1 in preventing transcription-replication conflicts (TRC) in murine muscle stem cells. They show that cells lacking Suv4-20h1 (SH1^{SKO} cells) exhibit reduced global H4K20me2 and an accumulation of H4K20me1 histone modifications. They correlate these modifications with increased transcription during S-phase and enhanced formation of TRC-induced R-loops and DNA damage in SH1^{SKO} cells. They show that loss of p53 in these cells overrides DNA-damage induced senescence and leads to malignant transformation and the development of rhabdomyosarcoma in mice. The authors also uncover that human sarcomas often harbour a low expression of SUV4-20H1 which correlates with poor patient outcome. The authors put forth a model in which SUV4-20H1 acts as a key tumour suppressor in sarcomas by repressing S-phase transcription, TRCs and associated DNA damage. The authors employ a variety of approaches to support their hypothesis, employing genome-wide analyses combined with cell biology assays and in vivo studies. The manuscript is well-structured and easily understood, although sometimes the figures are too data-dense. Below are some general points for considerations to the authors:

Response: We thank the reviewer for the positive evaluation of the manuscript. We have revised the manuscript following the reviewer’s recommendations and moved some nonessential data from main figures to supplemental figures.

1. The authors have well-designed experiments to answer their specific questions and put forth multiple interesting observations, however, some mechanistic links are missing and are worth re-considering. For example: the authors interpret observations in Suv4-20h1 KO as resulting from H4K20m1 increased levels. Although lack of Suv4-20h1 indeed results in higher H4K20m1 levels as demonstrated by the authors, other effects of this KO, not attributable to this histone mark, could not be excluded. It would be beneficial if the authors modulate the levels of other histone modifiers, such as SETD8 to control for some of their readouts, which would also serve as a control for the specificity of the role of Suv4-20h1. Alternatively, some complementation experiments with a catalytically dead Suv4-20h1 would provide the specificity of this enzymatic function to their observed phenotypes. In light of the authors using one cell model to support their findings, such control experiments are important.

Response: Following the reviewer’s suggestion, we monitored the protein levels of SETD8/KMT5A and the H4K20me1 demethylase PHF8 by western blot analysis in *Ctrl* and *Kmt5b^{SKO}* (SH1^{SKO}) MuSCs. Interestingly, protein levels of KMT5A decrease while levels of PHF8 increase after *Suv4-20h1* (*Kmt5b*) depletion, suggesting compensatory changes for H4K20me1 accumulation and indicating that the increase of H4K20me1 is the direct consequence of the absence of KMT5B (Revised Supplementary Fig. 2d).

Moreover, we also expressed human wildtype and a catalytical-inactive version of SUV4-20H1 (hKMT5B S251A) in *Ctrl* and *Kmt5b^{SKO}* MuSCs (revised Supplementary Fig. 2e). hKMT5B S251A carries a point mutation, which makes the enzyme catalytically inactive⁶. Expression of mutant hKMT5B resulted in dramatic depletion of H4K20me2 and strong accumulation of H4K20me1 in EdU+ *Ctrl* MuSCs (revised Extended Fig. 2f, g). We also detected increased numbers of PLA+ cells and enhanced accumulation of γ H2AX signals in EdU+ hKMT5B S251A expressing cells, which mimics the phenotype of *Kmt5b^{SKO}* MuSC, suggesting dominate negative

effects of the hKMT5B S251A mutant. Importantly, expression of wild-type but not the catalytically inactive version of human hKMT5B prevented all effects caused by inactivation of *Kmt5b* (revised Extended Fig. 2h, revised Supplementary Fig. 4b), demonstrating that loss of KMT5B-mediated conversion of H4K20me1 to H4K20me2 but not a non-enzymatic function of KMT5B is crucial for the phenotype.

2. The authors make a prominent point of how their findings in mouse cells apply to human sarcomas by extracting genomic data from available datasets and showing matching correlations (Fig. 7). While this is interesting, extrapolating their hypothesis to human cells requires at least recapitulating some of their main findings in human sarcoma cells. This is an important aspect for the suitability of the manuscript for a wider audience.

Response: We agree with the reviewer that it is important to validate the genomic data in human sarcoma cells, which was also asked for by reviewer #1 (comment #6). To comply with the reviewers' requests, we used different tumor cell lines to validate the findings from the TCGA analysis, including a rhabdomyosarcoma cell lines (RD), two carcinoma cell lines (MCF7, H1299), and an osteosarcoma cell line (U2OS). Two human myoblast cell lines and the human embryonic kidney cell line HEK293 were used as controls. We found a dramatic reduction of KMT5B expression and substantially higher levels of H4K20me1 in RD cells than the other cell lines (revised Supplementary Fig. 9b, c). Furthermore, we observed copy number loss and promoter hypermethylation of *KMT5B* in RD cells, explaining the lower expression of KMT5B (revised Supplementary Fig. 9d, e). In line with reduced KMT5B activity, RD cells displayed increased accumulation of PCNA-Pol II S2P PLA+ foci and R-loops, confirming the results of TCGA analysis, which link low activity of KMT5B with a high rate of transcription replication collision and sarcoma formation (revised Supplementary Fig. 9f, g).

3. It is important that the authors show cell cycle profiles of $SH1^{SKO}$ cells and if there are significant differences. A skew in cell cycle phase, for example, could account for observations the author report for whole-cell populations such as their western blots of DNA damage markers and histone modifications (Fig. 1g, Fig. 2a, Fig. 3d). In such cases, using synchronized cell populations and cell cycle-specific analyses will provide a more accurate representation of the effects observed.

Response: We agree with the reviewer that it would be helpful to synchronize cell populations when performing cell cycle-specific analyses. Unfortunately, we faced major technical challenges when trying to synchronize primary muscle stem cells. We have tried extensively to synchronize primary muscle stem cells using different chemicals such as nocodazole, thymidine and hydroxyurea. However, every time the cells all ended up with immediate differentiation after removal of the chemicals.

Instead, following the reviewer's suggestions, we performed FACS analysis to evaluate the effects of *Kmt5b* depletion on cell cycle progression by monitoring EdU incorporation and DAPI content (revised Supplementary Fig. 1h). Interestingly, the percentage of cells in the G0/G1- and S-phases was not significantly altered after inactivation of *Kmt5B*, although the percentage of cells in the G2/M-phase was slightly increased. We think that the minor increase of cells in the G2/M-phase is caused by aberrant mitoses following replication stress in mutant cells. To exclude the possibility that differences in cell cycle progression contribute to changes of histone modification and DNA damage markers (γ H2AX), we quantified the signals of H4K20me1, H4K20me2, H4K16ac, H3K27ac and γ H2AX in EdU+ and EdU- (excluding G2/M phase based on DAPI content) by

immunofluorescence staining (revised Fig. 1b, revised Supplementary Fig. 2a, revised Supplementary Fig. 3f, g). The results clearly demonstrate that the alterations of histone modifications and DNA damage markers in the whole cell population is predominately contributed by S-phase cells.

More specific comments with regards to the experiments:

4. Figure 1b: The authors use increases 53BP1 levels as a measure of DNA damage, although it is known that 53BP1 is recruited to DNA damage via H4K20me2 and a reduction in that mark would in fact decrease 53BP1 recruitment. The authors should address this discrepancy more carefully (line 335 they mention that there is no change in DNA damage response).

Response: We were also surprised to see the strong increase in 53BP1 foci in MuSC after inactivation of *Kmt5b* (revised Supplementary Fig. 1b, c), since it was reported that 53BP1 binds to H4K20me2 at DNA damage sites. However, the role of KMT5B/H4K20me2 in this process was not thoroughly characterized. Previous reports suggested H4K20me1 is sufficient whereas SUV4-20H1/H2 (KMT5A/5B) and H4K20me2 are dispensable for 53BP1 recruitment, since 53BP1 also binds to H4K20me1. Suv4-20h1/h2 DKO primary mouse embryonic fibroblasts (pMEFs), which show global reduction of H4K20me2 but elevation of H4K20me1, display only minor 53BP1 irradiation-induced foci (IRIF) defects⁷⁻¹⁰. In contrast, Suv4-20h1/h2 HeLa cells, which also show global reduction of H4K20me2 and increase of H4K20me1 are characterized by substantial 53BP1 IRIF defects^{11, 12}. In addition, Tuzon et al. demonstrated that PR-Set7-mediated H4K20me1 alone is insufficient for 53BP1 nucleation at induced double strand breaks (DSBs) and that SUV4-20H1/H2-mediated H4K20me2 formation is required¹³. Thus, the necessity of SUV4-20H1/KMT5B and/or H4K20me2 for recruitment of 53BP1 to DNA damage sites seems to be cell-type or context-dependent. It also seems possible that *Suv4-20h2* partially compensates for the function of *Suv4-20h1* and induces de novo H4K20me2 formation at DNA damage sites in *Suv4-20h1/Kmt5b* mutant MuSC. In support of this assumption, we observed retained H4K20me2 modifications in *Suv4-20h1/Kmt5b* depleted cells. In addition, overexpression of catalytic-inactive KMT5B in *Ctrl* MuSCs resulted in dominant negative effects, mimicking the *Kmt5b*^{SKO} phenotypes but without promoting 53BP1+ nuclear body formation. These results indicate that overexpression of catalytic-inactive KMT5B blocks recruitment of additional methyltransferases that may compensate for the function of KMT5B at DNA damage sites in *Kmt5b*^{SKO} MuSC (Figure 2 for the reviewer). Since a deep analysis of DNA damage responses is not the focus of this study, we changed the text and address this discrepancy more carefully in the discussion as suggested by the reviewer.

Figure 2 for the reviewer:

53BP1 nuclear bodies in *Ctrl* and *Kmt5b*^{SKO} MuSC in G1 phase (EdU- with 2N DNA content based on DAPI intensity) after transduction by empty overexpression vector, wildtype (WT) and catalytic-inactive (MUT) hKMT5B. Two-way ANOVA. n=3

5. Figure 2: The authors make a point about changing levels of H4K20me1 based on immunofluorescence images (Fig. 2c), but show no quantification or statistical analysis to back up these claims (also in supplementary data Fig. 2a,b,c). Similarly, in Figure 2c, it is not clear what threshold the authors used to quantify the cells as pATR-positive, as the images show definitely more than the 10% in the accompanied quantification. It is not clear why the analysis of p-RPA is not done in only EdU+ve cells, similar to other IF analyses as pRPA is also present at DNA double-strand breaks (and in G2 cells) and not only stalled forks. There is no quantification for this panel (Fig. 4d). In the case of DNA damage, γ H2AX foci quantification in multiple figures (Fig. 1g, Fig. 5a) in EdU+ve cells is a more sensitive, accurate measurement than total WB signal.

Response: We apologize for this shortcoming, which is also raised by reviewer 1 (comment #5) and reviewer 4 (comment #2). We assume the reviewer refers to the H4K20me1 and pATR staining in the old Fig. 2b and Fig. 4c. In the revised manuscript, we quantified the median fluorescence intensity (MFI) of each immunofluorescence staining from at least three biological replicates. The MFI of H4K20me1, H4K20me2, pATR, pRPA(s33) and γ H2AX signals in EdU+ /EdU- MuSCs was quantified and is shown in revised Fig. 2b, revised Supplementary Fig. 2a, revised Fig. 4d, e, Fig. 5b. The MFI of H4K20me3 in interphase, and H4K20me1/2/3 in metaphase was quantified and is shown next to the representative images in revised Supplementary Fig. 2b, c.

6. Figure 4: The PLA assay is really interesting and offers convincing data regarding the presence of TRC in SH1^{SKO} cells (Fig 4a). However, there are concerns regarding the specificity of the foci observed, especially that they are present in EdU-ve cells as well and if the authors have controlled for this (for example by depleting PCNA) or how do the authors explain the presence of PLA signal in these cells? Additionally, the authors also make a statement about PLA signal being found mainly in euchromatin (line 188), with no supporting data other than one IF image. Did the authors have quantify this?

Response: It would be nice to use PCNA-depleted MuSCs as control. Unfortunately, PCNA-depleted MuSC show reduced S-phase entry and rather undergo differentiation. Therefore, we employed alternative controls. We performed control PLA assays, in which either only the PCNA or only the Pol II S2P antibodies were used. Such negative controls have been described for the method and were used in published studies to monitor TRC¹⁴⁻¹⁶. In our case, the maximal number of foci from each single antibody assay was 3 (revised Supplementary Fig. 4a). Therefore, we only quantified cells with more than 3 foci to evaluate TRC in *Ctrl* and mutant cells. Standard and control for this method is described in the methods part. Only an average of 5.5~12.2 % PCNA-Pol II S2P PLA+/EdU- cells were detected without statistically significant differences between *Ctrl* and *Kmt5b*^{SKO} cells, indicating that the PLA assay is specific (Fig. 3 for the reviewer). It is worthwhile to mention that PCNA is recruited to DNA at G1/S transition prior to initiation of DNA synthesis^{17, 18}, which can cause additional PLA foci in EdU- cells.

We agree the reviewer that additional data are required to support the statement about localization of PLA signals mainly in the euchromatin. We now quantified the percentage of PLA foci in DAPI low and high regions, which is shown in the revised Fig. 4b, clearly supporting our conclusion that PLA signals are mainly found in DAPI-low euchromatin.

Figure 3 for the reviewer:

Quantification of PLA+ cells in EdU+/- *Ctrl* and *Kmt5b*^{SKO} MuSC.

(two-way ANOVA; n=3)

7. Figure 7: the authors make a point of increased ATR activation in sarcomas and compare it to other tumors but do not explain their choice of these tumors. Perhaps it is beneficial to include a group of tumors that are known to exhibit replication stress-associated ATR activation such as ovarian carcinomas, as positive control to their GSEA score. This is also predicted to be the case to at least for some BRCA-deficient breast cancer tumors, but it is not clear how the tumor data sets were selected and why no such increase is observed.

Response: We thank the reviewer for emphasizing the need to include a positive control for the ATR GSEA score. Following the reviewer's recommendation, we included the GSEA score of ATR in ovarian cancer (OV) (Fig. 4 for the reviewer). The median ATR score in SARC is slightly lower than in OV, but still higher than in all other cancers, supporting our claim of increased ATR activation in sarcomas. The median ATR in BRCA is lower than in SARC and OV, but still higher than in other cancers (Fig. 4 for the reviewer). We do not want to claim that ATR activation in response to replication stress is always associated with loss of KMT5B, but just want to compare the extend of ATR activation in SARC with other cancers showing higher levels of KMT5B.

It might be misleading to simply correlate the expression of KMT5B with ATR signaling pathway in different types of cancer, since loss of KMT5B is certainly not the only cause for activation of ATR in cancers. Changes in the genome of different cancer cells will have a major impact on the response to transcription replication collisions. Therefore, we have deleted the data on ATR/ATM signaling, moved all the human cancer data to Supplementary Fig. 8, and toned down our statements about the clinical implications of our findings in the discussion.

Figure 4 for the reviewer:

GSEA score of ATR activation in different cancers.

(one-way ANOVA)

Reviewer #3:

Previously, this group showed that KMT5B/SUV420H1 was necessary to maintain muscle satellite cell quiescence. They extend this finding in the current study to show that KMT5B/SUV420H1 is necessary to prevent S-phase transcription, transcription-replication-collisions (TRC), and tumor formation. In this study they show that deletion of KMT5B/SUV420H1 (which the authors abbreviate as SH1) in MuSCs results in signs of DNA double strand breaks (53BP1 foci, phospho-gH2AX foci, aberrant mitoses, increased p53 and p21) together with decreased cell expansion. This correlated with decreased H4K20me2 and increased H4K20me1 overall and specifically increased me1 in S-phase cells, which together with a correlation of increased me1 and PolII S2P in gene bodies and enhancer elements and increased EU incorporation in PCNA+ cells indicated that the enhanced transcription of these genes was present in S-phase. The authors use several approaches to show that this results in TRCs: proximity ligation assays (PLA) between PCNA and PolII together with EDU labeling, S9.6 Ab on dot-blots or together with MyoD antibody for IF or in ChIP, detection of phosphor ATR and RPA32, and reduction of gH2AX. PLA, and S9.6 foci with low dose DRB treatment. The analysis of the S9.6 ChIP showed enrichment of genes amplified or re-arranged in rhabdomyosarcoma and other cancers. Deletion of p53 rescued the proliferation defect of SH1 deficient cells and resulted in tumors during muscle regeneration in double-KOs (DKs), which had histologic characteristics of rhabdomyosarcomas. TCGA and other analyses showed prevalence of 11q and SH1 loss in sarcomas, as well as decreased SH1 and p53 RNA associated with elevated ATR and decreased ATM pathways.

Overall, this is a well done and well presented study.

The study convincingly shows increased S-phase transcription and TRCs with the acute total loss of SUV420H1/KMT5B, and tumor formation when coupled with acute loss of p53. The extension to human cancers has many important correlative findings that support a similar role in human sarcomas and other cancers. A major concern is that the current presentation of the data embraces the authors model and leaves the reader to identify additional considerations that might temper such an enthusiastic embrace. Although listed as major concerns below, these can mostly be addressed by changes in the text or minor experimental extensions.

Response: We thank the reviewer for the positive evaluation of the manuscript and for the constructive comments. We have revised the manuscript following the reviewer's suggestions.

Major concerns:

1. The mouse data nicely shows the TRCs and DNA damage associated with loss of KMT5B, as well as tumor formation when combined with p53 loss in a model of muscle regeneration. Genome instability, specifically re-arrangements and CNVs, is not directly shown in the resulting tumors and would strengthen the model. Showing this might be necessary to maintain the wording for the major conclusions. The title, abstract, and text makes the major conclusion that the de-repressed S-phase transcription causes genome instability. Is the demonstration of TRCs and gH2AX sufficient to conclude genome instability without a direct demonstration of re-arrangements or CNVs? Not for me, but I understand there might be some disagreement. Still, the understanding of the general reader should be considered and either wording for major conclusions changed and/or transparent definitions in the context of stating the major conclusions.

Response: We demonstrate high ratios of aneuploidy and micronuclei-containing *Kmt5b*^{SKO} and *Kmt5b/p53*^{SKO} MuSCs (revised Fig. 1e, f; revised Supplementary Fig. 7e, f). We now also

performed DNA fiber spreading assays to demonstrate that *Kmt5b* inactivation leads to stalled or slowed replication forks (revised Figure 1c), which is direct evidence for replication stress and genome instability of mutant MuSC. Furthermore, we performed qPCR-based copy number assays for cancer driver genes that gained R-loops in tumors derived from *Kmt5b/p53^{sKO}* MuSCs as suggested by the reviewer. Interestingly, tumor samples showed a gain for copy number of different oncogenes or a loss of tumor suppressors, demonstrating genome instability in the tumors (revised Supplementary Fig. 7h). In our view, the data conclusively demonstrate that TRC causes DNA under-replication and aberrant mitosis, which may contribute to genome instability including aneuploidy, micronuclei in *Kmt5b^{sKO}* and/or *Kmt5b/p53^{sKO}* MuSC, leading to copy number alteration in tumors.

2. The tumor formation in genetic models and in human data remains correlative. The authors nicely showed that loss of KMT5B resulted in TRCs, and that loss of KMT5B (or, in the case of human cancers, diminished KMT5B) was associated with tumors, but the study did not show that preventing TRCs in the face of KMT5B loss prevented the cancers and leaves open the possibility that other consequences of KMT5B loss might be the cause of transformation. The title and text is less nuanced and claims that de-repressed S-phase transcription causes malignant transformation. This concern can be addressed with either additional experimental studies, such as a rescue of tumor formation by preventing TRCs, or more careful wording.

Response: We agree with the reviewer's comment. It is theoretically possible that not only TRC but also other consequences of the loss of KMT5B may contribute to tumor formation. However, we demonstrated that attenuation of transcriptional elongation prevented several consequences of the loss of KMT5B using *in vitro* assays. In our view, this is a strong argument for the TRC hypothesis but indeed leaves room for interpretation. Due to the potential side effects of globally attenuated transcription caused by treatment DRB and the long process to obtain a permission for such animal experiments, we did not try to perform animal experiments *in vivo*. Instead, we toned down our statements and now use a more modest wording as suggested by the reviewer.

3. The generation of rhabdomyosarcoma, as opposed to other tumor types, in a mouse model that disrupts KMT5B in MuSCs seems directly related to the model system. To suggest specificity for RMS in the mouse model would require comparisons with deletions in other tissues at similar stages of tissue regeneration. Again, well beyond the scope of this study, but well within what needs to be more critically presented, particularly since this is used as a segue to human cancers and a focus on sarcomas.

Response: We thank the reviewer for raising this issue. Indeed, it is not possible for us to deplete *Suv4-20h1/Kmt5b* in numerous types of stem cells at different stages of tissue regeneration, analyze the function impact in different cell types and settings, and test for tumor formation. In the study, we focused on MuSCs and rhabdomyosarcoma. Although this approach limits the value of statements about other tumors and sarcomas, we think that some conclusions and generalizations are legitimate. We have deleted *Suv4-20h1/Kmt5b* in MEF cells but did not observe obvious DNA damage and proliferation defects in the mutant cells (Data not shown). Another study from our lab demonstrated that *Suv4-20h1* deficient ISL1+ cardiopulmonary progenitor cells and SM22+ smooth muscle cell/myofibroblast cells show a hyperproliferation phenotype¹⁹. These findings strongly indicate cell-type or context dependent functions of *Suv4-20h1/KMT5B*. Given that KMT5B is not the sole H4K20me2 methyltransferase, it is likely that compensatory actions of other

methyltransferases or demethylases account for the differences in various cell types. Expression of KMT5B is lowered more strongly in sarcomas compared to other cancer types, which is why we stressed the correlation of KMT5B expression with rhabdomyosarcoma in this study. It will be very interesting to investigate similarities and differences in the function of KMT5B and H4K20me1/2/3 in different cell types in the future and analyze how specificity is achieved. We have amended the discussion accordingly.

4. The mouse models relied on an acute complete deletion of KMT5B and did not assess heterozygotes, whereas the human cancers appear to have decreased expression but not necessarily complete loss. Does haploinsufficiency lead to TRCs? Maybe not necessary to experimentally address, but necessary to discuss.

Response: We thank the reviewer for raising this important point. We did not observe an obvious phenotype or tumor formation in heterozygous *Kmt5b* mutant but did observe dose-dependent effects in heterozygous *Kmt5b* mutant MuSCs (Fig. 5a for the reviewer). Heterozygous *Kmt5b* mutant MuSCs show a modest inhibition of proliferation and low-level of TRCs, which apparently is not sufficient to induce tumorigenesis in a *p53* knockout background (Fig. 5b, c for the reviewer).

Notably, 46.4% of sarcoma samples with loss of one *KMT5B* allele, 14.2% of sarcoma samples without copy number changes, and -surprisingly- even 13.8% of sarcoma samples with amplification of the *KMT5B* gene show comparable levels of *KMT5B* expression than sarcoma samples with mutations in *KMT5B*. These findings indicate that different mechanisms are involved in the reduction of *KMT5B* gene expression (revised Supplementary Fig. 8f). Our study indicates that DNA hypermethylation contributes to silencing of *KMT5B*, since we detected sarcoma-specific hyper-methylated regions proximal to the TSS (revised extended Fig. 8c). We assume that copy number loss and DNA hypermethylation in vicinity of the TSSs contribute to repression of *KMT5B* gene in SARCs, promoting tumorigenesis. We have discussed this issue in the revised manuscript.

Figure 5 for the reviewer:

- (a) RT-qPCR of *Kmt5b* in Ctrl, *Kmt5b*^{hetero} (*Pax7*^{CE+}/*Kmt5b*^{flax/+}) and *Kmt5b*^{SKO} MuSC.
- (b) Proliferation assay of Ctrl, *Kmt5b*^{hetero} (*Pax7*^{CE+}/*Kmt5b*^{flax/+}) and *Kmt5b*^{SKO} MuSC.
- (c) PLA assay of PCNA and Pol II S2P in Ctrl, *Kmt5b*^{hetero} (*Pax7*^{CE+}/*Kmt5b*^{flax/+}) and *Kmt5b*^{SKO} MuSC.

Data are shown as mean ± SEM; One-way ANOVA

5. Discussion of therapeutic implications in the discussion would be of interest. This might also address whether the TRC induced by *KMT5B* loss is transient or persistent in the mouse models, e.g., do the mouse tumors show continued TRC? It would also be interesting to address whether RMS cell lines show TRC that can be rescued by DRB. Again, maybe beyond the scope of this study, but a point that should be either addressed experimentally or in the Discussion.

Response: We thank the reviewer for raising this interesting question. As suggested, we performed PLA assays using anti-PCNA and anti-Pol II S2P in mouse tumor sections. Interestingly, 6 in 10 tumors from different *Kmt5b/p53^{sKO}* (DK) mice still showed TRCs, whereas the other four tumors were completely free of TRCs (revised Supplementary Fig. 7i, j). We speculate that tumors without TRCs have developed efficient TRC resolving mechanisms, such as via RNAP skipping, repriming, eviction and degradation, or fork cleavage and re-ligation²⁰. It is also possible that TRC- and non-TRC containing tumors differ in respect to the stage of tumor growth or immune cell infiltration pattern. We have not studied that in the current project but will do in the future.

In addition to mouse tumors, we also monitored the expression of *KMT5B* in human cells, a rhabdomyosarcoma cell line (RD), an osteosarcoma cell line (U2OS), two carcinoma cell lines (H1999, MCF-7), two human myoblast cell lines and a kidney cell line (HEK293). RT-qPCR results revealed strong reduction of *KMT5B* expression in RD cells due to copy number loss and promoter hypermethylation (revised Supplementary Fig. 9b, d, e). In line with reduced expression of *KMT5B*, RD cells displayed high numbers of PCNA-Pol II S2P PLA+ foci, which were reduced in a dose-dependent manner by treatment with DRB (revised Supplementary Fig. 9f, h). However, proliferation of RD cells was not affected by DRB treatment, which argues against the therapeutic potential of DRB or similar compounds for the treatment of *KMT5B*-low tumors (revised Supplementary Fig. 9i). Low level of *KMT5B* can be used as a prognostic marker, since its expression negatively correlates with the risk of tumor relapse (revised Supplementary Fig. 8b). We have toned down the statements about potential clinical implications, since we did not go deep enough into this matter with the current study, which primarily focuses on muscle stem cells and the mechanisms involved in malignant transformation.

6. Figure 7A shows a heat plot for the methylation beta value where a higher number should reflect a higher percentage of methylation. The figure shows lower beta values in the promoter region of sarcomas, the text indicates that these lower values mean higher methylation; whereas panels e and f show higher values for the sarcomas (not specified whether these are beta values) indicating higher methylation. Maybe I am misunderstanding, but something seems inconsistent.

Response: We are sorry for the confusion and carefully checked the heat map data. Higher beta values as shown in the original Fig. 7a, e, f indicate higher methylation. The regions with lower beta values in SARC described by the reviewer are not annotated as promoter regions. The sarcoma-specific hyper-methylated region with higher beta values (marked by numbers in the revised Supplementary Fig. 8c) is located in a promoter-associated region (supplementary table 3). This promoter associated region is proximal to the TSS (approximately ± 1 kb), where hypermethylation of tumor suppressors in tumors is associated with transcriptional repression²¹.

The DNA methylation levels in original Fig. 7e, f are represented by beta values as well. We have indicated this in the legend of the revised figure (revised Supplementary Fig. 8 h, i). We apologize that the imperfect description caused confusion.

Minor points:

7. *Should the authors use the official gene and protein name, KMT5B?*

Response: The gene name *Suv4-20h1* is still broadly used but the reviewer is right that the official gene name should be used. We changed to the official gene and protein names in the revised figures and in the manuscript, mentioning in the beginning that the *Kmt5b* gene is also known as *Suv4-20h1*.

8. *Page 8 introduces the term TCR as compared to TRC, likely an error?*

Response: Indeed, this was a typo, which has been corrected. Thank you for the careful reading.

9. Page 11 line 285 classifies the mouse tumors as embryonal rhabdomyosarcoma. This seems to go beyond the data or even the ability to classify genetic mouse tumors with a human classification system.

Response: We agree with the reviewer. The specification ‘embryonal’ has been removed.

Reviewer #4

The manuscript 'De-repressed S-phase transcription induces replication collisions causing genomic instability and malignant transformation of adult stem cells' by Zhang et al., investigates the effects of Suv4-20h1 loss on transcription-replication conflicts in adult muscle stem cells. This is generally an excellent, well-thought-out and executed study with wide-ranging implication for the general understanding of how transcription-replication conflicts occur and the potential involvement of Suv4-20h1 in the formation of rhabdomyosarcomas.

Despite my sentiments I do have some minor and major comments that in my opinion are essential before publication.

Response: We thank the reviewer for the positive evaluation of the manuscript and for the constructive comments.

Major

1. On page 5 lines 100-101 the authors write: 'We concluded that loss of Suv4-20h1 causes DNA damage and replication stress in proliferating SH1^{SKO} MuSCs both in vitro and in vivo.' Increased 53BP1, γ H2AX, micronuclei and aneuploidy imply replication stress however replication stress is defined by DNA synthesis slowing and/or replication fork stalling (PMID: 25907220). Therefore, the author must show DNA synthesis slowing and/or replication fork stalling using the DNA Fibre assay in MuSC cells. Furthermore, this statement would be better suited at the end of the next paragraph (circa lines 112/113)

Response: We thank the reviewer for this helpful suggestion. We performed DNA fiber assays using CldU/IdU for pulse labeling. Reduction of IdU tract lengths and increased number of fibers retaining only the first label (CldU) indicate that depletion of *Kmt5b* causes stalling of the replication fork (revised Fig. 1c, d). The new results support our conclusion that *Kmt5b* inactivation provokes replication stress.

As suggested by the reviewer, we moved the statement 'we concluded that loss of *Kmt5b* causes DNA damage, replication stress, and genome instability in proliferating *Kmt5b*^{SKO} MuSCs both *in vitro* and *in vivo* .' to the end of the next paragraph.

2. On page 6 lines 128-130 the authors discuss Genome-wide deposition of H4K20me1 shown in figure 2B. These images only show several individual cells without any quantification and thus do not provide any evidence for the H4K20me1 accumulates in S-phase. The author must provide quantification of EdU (+/-) and H4K20me1 (+/-) similar to data shown in figure 3E. In line with these changes the authors must modify the discussion accordingly if this data is not provided (page 12 lines 331-332).

Response: We apologize for this shortcoming, which was also recognized by the other reviewers. As explained in our responses to comment #5 of reviewer #1 and comment #5 of reviewer #2, we quantified the median fluorescent intensity (MFI) of H4K20me1 in EdU+/EdU- *Ctrl* and *Kmt5b*^{SKO} MuSCs from three biological replicates. The results support the conclusion that loss of *Kmt5b* leads to accumulation of H4K20me1 predominantly in S-phase (revised Fig. 1b).

3. The data shown in Figure 5 adequately demonstrates repression of transcriptional elongation reduces R-loop formation (and potentially genomic instability). However, the authors have provide

little evidence for the effects on transcription-replication conflicts. In figure 5A, the γ H2AX quantification and western blots do not correlated and are not convincing. The authors need to repeat experiments shown in Fig 1B-F looking at either 53BP1 or γ H2AX in EdU+ cells +/- DRB.

Response: We thank the reviewer for raising this important point. We repeated the experiments for aneuploidy and micronuclei analysis after DRB treatment as suggested by the reviewer. The results indicate that repression of transcription elongation ameliorates increased aneuploidy and micronuclei caused by deletion of *Kmt5b*, demonstrating that transcription replication conflicts are a major cause of genome instability in *Kmt5b*^{SKO} MuSC (revised Supplementary Fig. 6b, c).

We also quantified the intensity of immunofluorescence staining for γ H2AX in EdU+ cells after PBS or DRB treatment using more biological replicates (n=4). Consistent with the results from the western blot analysis, DRB treatment dramatically reduced the signal strength of γ H2AX in *Kmt5b*^{SKO} MuSCs (revised Fig. 5b). We had also obtained statistically significant results for the western blot analysis of γ H2AX in the DRB experiment. We agree that the previous representative western blot images did not accurately reflect the quantification (*Ctrl* n=3; *Kmt5b*^{SKO}: n=4). Therefore, we repeated the experiment and increased the number of replicates for the γ H2AX western blot analysis (*Ctrl* n=5; *Kmt5b*^{SKO}: n=6). The quantification of western blot analysis revealed significant differences (without DRB), respectively the lack of it (with DRB), which is consistent with the immunofluorescence staining. Since we included the quantification of γ H2AX immunofluorescence intensity in EdU+ cells after PBS and DRB treatment in the revised Fig. 5b (main figure), we moved the extended western blot analysis into the revised supplementary Fig. 6a. Taken together, we confirmed statistically significant differences of γ H2AX signals between control and mutant cells using two different methods.

4. I have major reservation about the interpretation of the R-loop data in this manuscript. On page 8 lines 190-191 the authors write: 'In case of head-on²² orientations, TRCs replication forks will stall and promote formation of genome-destabilizing R-loop.' Firstly, the grammar in this sentence needs addressing. Secondly the authors here and in accompanying statements throughout the manuscript, are suggesting head-on²² transcriptional-replication conflicts cause R-loop formation. This may well be true however R-loops can be generated in numerous ways either physiologically (e.g. transcription or DNA damage repair) and pathologically (excessive transcription following oncogene activation). The experiments discussed in fig 4 or Supplementary Fig. 4 do not demonstrate head-on-collisions between replication or transcription machinery. The data only shows increased R-loop formation following the loss of *SUV420H1*. On page 8 lines 204-205 the authors also write: 'To determine precisely the locations of TCR-induced R-loops' Again the authors have not provided any evidence of the 'type' of transcription-replication conflicts or R-loops were generated following TRC. Therefore, the authors are over stating their findings (also in the discussion) based on a false premise.

The authors need to delete any reference to head-on-collisions or that R-loops are generated following collision of the replisome and transcriptional machinery. For example the following statement must be amended (page 10 lines 249-252): 'Taken together, these data indicate that increased transcription activity of a large group of coding genes and eRNAs during S-phase promotes collision of the replisome and transcriptional machinery leading to R-loop formation and genome instability of *SH1*^{SKO} MuSC (Fig.5g).'

On page 8/9 line 209-212, the authors contradict their previous premise: '64.2% (1508 from 2350) of R-loop containing genes showed increased *H4K20me1* and Pol II S2P enrichment, indicating

that enhanced transcriptional activation is the main cause for increased R-loop formation in *SH1^{SKO} MuSC*. 'Are R-loops caused by head-on-collisions or enhanced transcriptional activation? In my opinion the latter, since 64.2% (1508 from 2350) of R-loop containing genes showed increased H4K20me1 and Pol II S2P enrichment and DRB treatment reduces R-loop formation! Finally, in order to show Increased R-loop formation causes transcription-replication conflicts the authors would be required to assess DNA damage (RPA, 53BP1 or γ H2AX in EdU+ cells at a minimum) in cells with or without RNaseH1 (see PMID: 27725641).

Response: We agree with the reviewer that we do not provide direct evidence for the orientation of transcription replication collisions. In theory, S-phase transcription may result in either head-on or codirectional TRCs. Several papers have claimed that head-on collisions between RNAPs and the replication machinery are stable, whereas co-directional collisions are relatively transient. Since the collision foci formed in early replicating regions of *Kmt5^{SKO} MuSCs* (revised Fig. 4a) were still unresolved in late S-phase, we hypothesize that we observed long-lasting, stable head-on collisions in *Kmt5^{SKO} MuSC*. Of course, this is not a definitive proof and also does not exclude the possibility of co-directional collisions. We have modified our statements in the revised manuscript and are not referring to head-on-collisions anymore with the exception of a single paragraph in the discussion, where we also clearly point out the limitations of the analysis.

We politely disagree with the reviewer's view that the conclusion 'enhanced transcriptional activation is the main cause for increased R-loop formation in *Kmt5b^{SKO} MuSC*' contradicts the previous premise that TRCs cause R-loop formation in *Kmt5b*-deficient MuSCs. We provide clear evidence that the increase of H4K20me1, caused by the absence of *Kmt5b*, results in increased transcriptional activation, particularly during S-phase, which is indispensable, albeit not sufficient, for formation of unresolvable R-loops. Several lines of evidence support the causal effect of TRCs, as consequence of elevated S-phase transcriptional activity, in *Kmt5b^{SKO} MuSCs* on unresolved R-loop accumulation: (i) only 1508 genes out of 8020 transcriptionally upregulated genes showed R-loop formation at gene bodies, suggesting that transcriptional activation alone is not sufficient to cause formation of stable R-loops (revised Fig. 4g). (ii) using purified genomic DNA from FACS-isolated S- and G1-phase MuSCs, we demonstrate that R-loops mainly accumulate during S-phase rather than in G1-phase, indicating that increased R-loops are associated with DNA replication (revised Supplementary Fig. 4d). (iii) genomic mapping of R-loops by DRIP-seq revealed strong accumulation of aberrant R-loops mainly in gene bodies but not at TSS and TES, where R-loops involved in transcriptional regulation normally localize to exert their function for transcription pausing and termination.

To further address the question whether R-loops are the cause or the consequence of TRCs, we performed RNaseH1 overexpression experiments as suggested by the reviewer (revised Supplementary Fig. 6e). Interestingly, increased expression of mRNASH1 led to a marked reduction of R-loop formation in both control and *Kmt5b^{SKO} MuSCs* (revised Fig. 5f), but the levels of PCNA-Pol II S2P PLA+ foci, γ H2AX and p-RPA in EdU+ *Kmt5b^{SKO} MuSC* were only mildly affected compared to control-transduced *Kmt5b^{SKO} MuSC* (revised Fig. 5g, h, revised Supplementary Fig. 6f). These findings indicate that TRC-induced genomic damage does only depend to a minor degree on R-loop formation. On the other hand, DRB treatment completely prevented TRCs, R-loop formation and γ H2AX accumulation in *Kmt5b*-deficient MuSCs (revised Fig. 5b-d). In sum, the new data provide additional evidence that TRCs cause formation of aberrant R-loops in *Kmt5b^{SKO} MuSCs*. In fact, R-loops without RNAP only serve as a transient obstacle to replication fork progression, while the complex resulting from head-on collisions serves as more severe and stable road block²³.

Taken together, we sincerely think that the reviewer's question "Are R-loops caused by head-on-collisions or enhanced transcriptional activation?" is somewhat misleading. We are convinced that we provided ample evidence that R-loops are caused by transcription-replication collisions resulting from enhanced transcriptional activation, particularly during S-phase. We also hope the reviewer agrees that the new results support our conclusion about the order of events: first enhanced transcriptional activation, primarily during S-phase, due to accumulation of H4K20h1, next TRCs, and then R-loop formation.

To improve the presentation of our findings, we have changed the wording in the manuscript. We do not make conclusions anymore about the orientation of TRCs. Instead, we discuss the causality between TRCs and R-loops.

Minor

1. *in the introduction on page 3 line 65, the abbreviation MuSC needs defining.*

Response: MuSC has been defined. Thank you for the careful reading.

2. *On page 5 line 97 the authors write: 'Importantly, a marked increase of 53BP1 nuclear bodies¹ was apparent in the G1 phase of SH1^{SKO} MuSC, suggesting failure to complete DNA replication during the last cell cycle (Fig.1c).' It is not evident how the G1-phase or 2N DNA content was identified. Authors must provide this information.*

Response: Cells at G1 phase were identified as EdU- containing 2N DNA content based on DAPI intensity. The information has been provided in the figure legend in revised Supplementary Fig. 1c.

3. *In the methods on page 17 line 481 the title '9.6 Dot blot; needs changing to S9.6 Dot blot.*

Response: The text has been corrected. Thank you.

4. *On page 9 line 224, Figure references needs amending.*

Response: Figure references have been amended.

5. *On page 9 line 236, the author must provide details of how long cells were treated with DRB.*

Response: The cells was treated with DRB 2 days after seeding for 48 hours. This information is included in Method part (MuSC purification and culture) in the revised manuscript.

6. *On page 10 CTX abbreviation needs defining.*

Response: CTX stands for cardiotoxin. We have introduced a definition in line 102, page 5, where CTX was mentioned for the first time

7. *On page 14 the sentence on lines 371-372 needs some context.*

Response: The sentence has been changed.

8. The y-axis in Fig 6G needs amending.

Response: The y-axis has been corrected.

REFERENCES

1. Beck, D.B., Oda, H., Shen, S.S. & Reinberg, D. PR-Set7 and H4K20me1: at the crossroads of genome integrity, cell cycle, chromosome condensation, and transcription. *Genes Dev* **26**, 325-337 (2012).
2. Pesavento, J.J., Yang, H., Kelleher, N.L. & Mizzen, C.A. Certain and progressive methylation of histone H4 at lysine 20 during the cell cycle. *Mol Cell Biol* **28**, 468-486 (2008).
3. Svensson, J.P. *et al.* A nucleosome turnover map reveals that the stability of histone H4 Lys20 methylation depends on histone recycling in transcribed chromatin. *Genome Res* **25**, 872-883 (2015).
4. Nakamura, K. *et al.* H4K20me0 recognition by BRCA1-BARD1 directs homologous recombination to sister chromatids. *Nat Cell Biol* **21**, 311-318 (2019).
5. Pellegrino, S., Michelena, J., Teloni, F., Imhof, R. & Altmeyer, M. Replication-Coupled Dilution of H4K20me2 Guides 53BP1 to Pre-replicative Chromatin. *Cell Rep* **19**, 1819-1831 (2017).
6. Wu, H. *et al.* Crystal structures of the human histone H4K20 methyltransferases SUV420H1 and SUV420H2. *FEBS Lett* **587**, 3859-3868 (2013).
7. Botuyan, M.V. *et al.* Structural basis for the methylation state-specific recognition of histone H4-K20 by 53BP1 and Crb2 in DNA repair. *Cell* **127**, 1361-1373 (2006).
8. Hartlerode, A.J. *et al.* Impact of histone H4 lysine 20 methylation on 53BP1 responses to chromosomal double strand breaks. *PLoS One* **7**, e49211 (2012).
9. Oda, H. *et al.* Regulation of the histone H4 monomethylase PR-Set7 by CRL4(Cdt2)-mediated PCNA-dependent degradation during DNA damage. *Mol Cell* **40**, 364-376 (2010).
10. Schotta, G. *et al.* A chromatin-wide transition to H4K20 monomethylation impairs genome integrity and programmed DNA rearrangements in the mouse. *Genes Dev* **22**, 2048-2061 (2008).
11. Hsiao, K.Y. & Mizzen, C.A. Histone H4 deacetylation facilitates 53BP1 DNA damage signaling and double-strand break repair. *J Mol Cell Biol* **5**, 157-165 (2013).
12. Yang, H. *et al.* Preferential dimethylation of histone H4 lysine 20 by Suv4-20. *J Biol Chem* **283**, 12085-12092 (2008).
13. Tuzon, C.T. *et al.* Concerted activities of distinct H4K20 methyltransferases at DNA double-strand breaks regulate 53BP1 nucleation and NHEJ-directed repair. *Cell Rep* **8**, 430-438 (2014).
14. Hamperl, S., Bocek, M.J., Saldivar, J.C., Swigut, T. & Cimprich, K.A. Transcription-Replication Conflict Orientation Modulates R-Loop Levels and Activates Distinct DNA Damage Responses. *Cell* **170**, 774-786 e719 (2017).
15. Kim, S. *et al.* ATAD5 restricts R-loop formation through PCNA unloading and RNA helicase maintenance at the replication fork. *Nucleic Acids Res* **48**, 7218-7238 (2020).
16. Tsai, S. *et al.* ARID1A regulates R-loop associated DNA replication stress. *PLoS Genet* **17**, e1009238 (2021).
17. Mukherjee, P., Cao, T.V., Winter, S.L. & Alexandrow, M.G. Mammalian MCM loading in late-G(1) coincides with Rb hyperphosphorylation and the transition to post-transcriptional control of progression into S-phase. *PLoS One* **4**, e5462 (2009).
18. Takahashi, T. & Caviness, V.S., Jr. PCNA-binding to DNA at the G1/S transition in proliferating cells of the developing cerebral wall. *J Neurocytol* **22**, 1096-1102 (1993).
19. Qi, H. *et al.* Epigenetic Regulation by Suv4-20h1 in Cardiopulmonary Progenitor Cells Is Required to Prevent Pulmonary Hypertension and Chronic Obstructive Pulmonary Disease. *Circulation* **144**, 1042-1058 (2021).
20. Lalonde, M., Trauner, M., Werner, M. & Hamperl, S. Consequences and Resolution of Transcription-Replication Conflicts. *Life (Basel)* **11** (2021).
21. Ando, M. *et al.* Chromatin dysregulation and DNA methylation at transcription start sites associated with transcriptional repression in cancers. *Nat Commun* **10**, 2188 (2019).
22. Liu, W. *et al.* PHF8 mediates histone H4 lysine 20 demethylation events involved in cell cycle progression. *Nature* **466**, 508-512 (2010).

23. Bruning, J.G. & Marians, K.J. Replisome bypass of transcription complexes and R-loops. *Nucleic Acids Res* **48**, 10353-10367 (2020).

REVIEWERS' COMMENTS

Reviewer #1 (Remarks to the Author):

The authors have satisfactorily addressed my concerns.

Reviewer #2 (Remarks to the Author):

I thank the authors for their efforts to carefully and thoroughly address the raised concerns and comments to more convincingly support their claims. They added essential controls for some of the methods, such as the PLA and the add-back of hKMT5B, which I found strengthened the conclusions from these experiments. Additional quantification of IF images and cell-cycle specific analyses also highlighted and corroborated the S-phase specificity of their observations. The authors also adjusted their arguments regarding the applicability of their model to human cells and added multiple cell lines to build on their narrative in this section. Finally, they accordingly revised the structure and wording of the manuscript to highlight the strengths of the study and to avoid over-interpretation of some findings. I am satisfied with their responses and explanations and the data they have provided in the rebuttal (which are not necessary to add to the manuscript). I find that the manuscript has overall considerably improved in quality and clarity and is now suitable for publication.

Reviewer #3 (Remarks to the Author):

The authors have addressed my prior critiques to my satisfaction.

Reviewer #4 (Remarks to the Author):

The revised manuscript 'Replication collisions induced by de-repressed S-phase transcription are connected with malignant transformation of adult stem cells' has addressed all the issues highlighted by all the reviewers. These changes in my opinion significantly improve the manuscript and our mechanistic understanding of the potential involvement of KMT5B in the formation of rhabdomyosarcomas. Moreover it provides new insights in to the causes of TRC and genomic instability. Thus I highly recommend this excellent study for publication.

I only found a few minor issues:

- 1) On line 102, Fig 1d is incorrectly referenced.
- 2) On lines 345 and 346, the authors write: 'Of note, a large group of sarcomas showed low expression of both KMT5B and TP53 (Supplementary Fig.8c, i).' Supplementary Figure 8i refers to DNA methylation, should this be changed to Supplementary Figure 8j?
- 3) Again on line 349, Supplementary Fig.8l is referred to however I assume this is should be Supplementary Fig.8k!

Response to reviewers

Reviewer #1 (Remarks to the Author):

The authors have satisfactorily addressed my concerns.

Response: We are delighted about the positive reception of the revised paper.

Reviewer #2 (Remarks to the Author):

I thank the authors for their efforts to carefully and thoroughly address the raised concerns and comments to more convincingly support their claims. They added essential controls for some of the methods, such as the PLA and the add-back of hKMT5B, which I found strengthened the conclusions from these experiments. Additional quantification of IF images and cell-cycle specific analyses also highlighted and corroborated the S-phase specificity of their observations. The authors also adjusted their arguments regarding the applicability of their model to human cells and added multiple cell lines to build on their narrative in this section. Finally, they accordingly revised the structure and wording of the manuscript to highlight the strengths of the study and to avoid over-interpretation of some findings. I am satisfied with their responses and explanations and the data they have provided in the rebuttal (which are not necessary to add to the manuscript). I find that the manuscript has overall considerably improved in quality and clarity and is now suitable for publication.

Response: We appreciate the supporting comments by the reviewer and are glad that we addressed all open issues.

Reviewer #3 (Remarks to the Author):

The authors have addressed my prior critiques to my satisfaction.

Response: We thank the reviewer for the positive comments.

Reviewer #4 (Remarks to the Author):

The revised manuscript ‘Replication collisions induced by de-repressed S-phase transcription are connected with malignant transformation of adult stem cells’ has addressed all the issues highlighted by all the reviewers. These changes in my opinion significantly improve the manuscript and our mechanistic understanding of the potential involvement of KMT5B in the formation of rhabdomyosarcomas. Moreover it provides new insights in to the causes of TRC and genomic instability. Thus I highly recommend this excellent study for publication.

I only found a few minor issues:

1) On line 102, Fig 1d is incorrectly referenced. 2) On lines 345 and 346, the authors write: ‘Of note, a large group of sarcomas showed low expression of both KMT5B and TP53 (Supplementary Fig.8c, i).’

Supplementary Figure 8i refers to DNA methylation, should this be changed to Supplementary Figure 8j?
3) Again on line 349, Supplementary Fig.8l is referred to however I assume this is should be Supplementary Fig.8k!

Response: We apologize for the mistakes and thank the reviewer for the careful reading. We have corrected the manuscript accordingly.

- 1) The incorrect referencing to (Fig. 1d) in line 102 was removed.
- 2) (Supplementary Fig.8c, i) in line 346 was changed to (Supplementary Fig.8c, j).
- 3) (Supplementary Fig.8l) in line 349 was been changed to (Supplementary Fig.8k).